# Model-Based Multi-Agent RL in Zero-Sum Markov Games with Near-Optimal Sample Complexity

**Kaiqing Zhang**
ECE and CSL
University of Illinois at Urbana-Champaign
kzhang66@illinois.edu

**Sham M. Kakade**
CS and Statistics
University of Washington
Microsoft Research
sham@cs.washington.edu

**Tamer Başar**
ECE and CSL
University of Illinois at Urbana-Champaign
basar1@illinois.edu

**Lin F. Yang**
ECE
University of California, Los Angeles
linyang@ee.ucla.edu

## Abstract

Model-based reinforcement learning (RL), which finds an optimal policy using an empirical model, has long been recognized as one of the cornerstones of RL. It is especially suitable for multi-agent RL (MARL), as it naturally decouples the *learning* and the *planning* phases, and avoids the *non-stationarity* problem when all agents are improving their policies simultaneously using samples. Though intuitive and widely-used, the sample complexity of model-based MARL algorithms has been investigated relatively much less often. In this paper, we aim to address the fundamental open question about the sample complexity of model-based MARL. We study arguably the most basic MARL setting: two-player discounted zero-sum Markov games, given only access to a generative model of state transition. We show that model-based MARL achieves a sample complexity of $\widetilde{\mathcal{O}}(|\mathcal{S}||\mathcal{A}||\mathcal{B}|(1-\gamma)^{-3}\epsilon^{-2})$ for finding the Nash equilibrium (NE) *value* up to some $\epsilon$ error, and the $\epsilon$-NE *policies*, where $\gamma$ is the discount factor, and $\mathcal{S}, \mathcal{A}, \mathcal{B}$ denote the state space, and the action spaces for the two agents. We also show that this method is near-minimax optimal with a tight dependence on $1 - \gamma$ and $|\mathcal{S}|$ by providing a lower bound of $\Omega(|\mathcal{S}|(|\mathcal{A}| + |\mathcal{B}|)(1-\gamma)^{-3}\epsilon^{-2})$. Our results justify the efficiency of this simple model-based approach in the multi-agent RL setting.

## 1 Introduction

Recent years have witnessed phenomenal successes of reinforcement learning (RL) in many applications, e.g., playing strategy games [1, 2], playing the game of Go [3, 4], autonomous driving [5], and security [6, 7]. Most of these successful while practical applications involve more than one decision-maker, giving birth to the surging interests and efforts in studying multi-agent RL (MARL) recently, especially on the theoretical side [8, 9, 10, 11, 12, 13]. See also comprehensive surveys on MARL in [14, 15, 16].

In general MARL, all agents affect both the state transition and the rewards of each other, while each agent may possess different, sometimes even totally conflicting objectives. Without knowledge of the model, the agents have to resort to data to either estimate the model, improve their own policy, and/or infer other agents' policies. One fundamental challenge in MARL is the emergence of *non-stationarity* during the learning process [14, 15]: when multiple agents improve their policies concurrently and directly using samples, the environment becomes non-stationary from each agent's

perspective. This has posed great challenge to development of effective MARL algorithms based on single-agent ones, especially *model-free* ones, as the condition for guaranteeing convergence in the latter fails to hold in MARL, and additional non-trivial efforts are required to address it [17, 18, 19]. One tempting remedy is the simple while intuitive method – model-based MARL: one first estimates an empirical model using data, and then finds the optimal, more specifically, equilibrium policies in this empirical model, via planning. Model-based MARL naturally decouples the *learning* and *planning* phases, and can be incorporated with *any* black-box planning algorithm that is efficient, e.g., value iteration [20] and (generalized) policy iteration [21, 22].

Though intuitive and widely-used, rigorous theoretical justifications for these model-based MARL methods are relatively rare. In this work, we aim to answer the following standing question: how good is the performance of this simple "plug-in" method in terms of non-asymptotic sample complexity? To this end, we focus on arguably the most basic setting of MARL, well recognized ever since [17]: two-player discounted zero-sum Markov games (MGs) with simultaneous-move agents, given only access to a generative model of state transition. This generative model allows agents to sample the MG, and query the next state from the transition process, given any state-action pair as input. The generative model setting has been a benchmark in RL when studying the sample efficiency of algorithms [23, 24, 25, 26, 27]. Indeed, this model allows the study of sample-based multi-agent planning over a long horizon, and helps develop better understanding of the statistical properties of the algorithms, decoupled from the exploration complexity.

Motivated by recent minimax optimal complexity results for single-agent model-based RL [23], we address the question above with a positive answer: model-based MARL approach can achieve near-minimax optimal sample complexity in finding both the Nash equilibrium (NE) value and policies. See a detailed description as follows. Our results have justified the efficiency of this simple model-based approach to MARL.

**Contribution.** We establish the sample complexities of model-based MARL in zero-sum Markov games, when a generative model is available. We show that the approximate solution to the empirical model can achieve the NE value of the true model up to some $\epsilon$-error with $\widetilde{\mathcal{O}}(|\mathcal{S}||\mathcal{A}||\mathcal{B}|(1-\gamma)^{-3}\epsilon^{-2})$ samples, where $\widetilde{\mathcal{O}}$ suppresses the logarithmic factors, $\gamma$ is the discount factor, and $\mathcal{S}, \mathcal{A}, \mathcal{B}$ denote the state space, and the action spaces for the two agents, respectively. Establishing a lower bound of $\Omega(|\mathcal{S}|(|\mathcal{A}| + |\mathcal{B}|)(1-\gamma)^{-3}\epsilon^{-2})$, we show that this simple method is indeed near-minimax optimal, with a tight dependence on $1 - \gamma$ and $|\mathcal{S}|$, and a sublinear dependence on model-size. This result then induces a $\widetilde{\mathcal{O}}(|\mathcal{S}||\mathcal{A}||\mathcal{B}|(1-\gamma)^{-5}\epsilon^{-2})$ sample complexity for achieving $\epsilon$-NE policies. Moreover, we provide a planning oracle that is *smooth* in producing the approximate NE policies, and show that obtaining $\epsilon$-NE policies can also achieve the near-optimal complexity of $\widetilde{\mathcal{O}}(|\mathcal{S}||\mathcal{A}||\mathcal{B}|(1-\gamma)^{-3}\epsilon^{-2})$. These near-optimal results are first-of-their-kind in model-based MARL, to the best of our knowledge.

**Related Work.** Stemming from the seminal work [17], MARL has been mostly studied under the framework of Markov games [20]. There is no shortage of provably convergent MARL algorithms ever since then [19, 18, 28]. However, most of these early results are Q-learning-based (thus model-free) and asymptotic, with no sample complexity guarantees. To establish finite-sample results, [22, 29, 30, 31, 32] have studied the sample complexity of batch model-free MARL methods. There are also increasing interests in policy-based (thus also model-free) methods for solving special MGs with non-asymptotic convergence guarantees [33, 34, 11]. No result on the (near-)minimax optimality of these complexities has been established.

Specific to the two-player zero-sum setting, [35, 10] have considered *turn-based MGs*, a special case of the simultaneous-move MGs considered here, with a generative model. Specifically, [10] established near-optimal sample complexity of $\widetilde{\mathcal{O}}((1-\gamma)^{-3}\epsilon^{-2})$ for a variant of Q-learning for this setting. More recently, [36, 12] have established both regret and sample complexity guarantees for finite-horizon zero-sum MGs, without a generative model, with focus on efficient exploration. The work in [13] also focused on the turn-based setting, and combined Monte-Carlo Tree Search and supervised learning to find the NE values. In contrast, model-based MARL theory has a relatively limited literature. [37] proposed the R-MAX algorithm for average-reward MGs, with polynomial sample complexity. [8] developed a model-based upper confidence algorithm with polynomial sample complexities for the same setting. These methods differ from ours, as they are either model-free approaches, or not clear yet if they are (near-)minimax optimal in the corresponding setups.

In the single-agent regime, there has been extensive literature on non-asymptotic efficiency of RL in MDPs; see [24, 25, 38, 39, 26, 40, 41, 42, 43, 27, 44]. Amongst them, we highlight the minimax optimal ones: [26] and [42] have provided minimax optimal results for sample complexity and regret in the settings with and without a generative model, respectively. Specifically, [26] has shown that to achieve the $\epsilon$-optimal *value* in Markov decision processes (MDPs), at least $\Omega(|\mathcal{S}||\mathcal{A}|(1-\gamma)^{-3}\epsilon^{-2})$ samples are needed, for $\epsilon \in (0, 1]$. They also showed that to find an $\epsilon$-optimal *policy*, the same minimax complexity order in $1-\gamma$ and $\epsilon$ can be attained, if $\epsilon \in (0, (1-\gamma)^{-1/2}|\mathcal{S}|^{-1/2}]$ and the total sample complexity is $\widetilde{\mathcal{O}}(|\mathcal{S}|^2|\mathcal{A}|)$, which is in fact *linear* in the model size. Later, [27] has proposed a Q-learning based approach to attain this lower bound and remove the extra dependence on $|\mathcal{S}|$, for $\epsilon \in (0, 1]$. More recently, [23] developed new techniques based on *absorbing MDPs*, to show that model-based RL also achieves the lower bound for finding an $\epsilon$-optimal *policy*, with a larger $\epsilon$ range of $(0, (1-\gamma)^{-1/2}]$, which has motivated our present work. While preparing the present work, [45] has further improved the results in [23], in that they cover the entire range of sample sizes.

## 2   Preliminaries

**Zero-Sum Markov Games**   Consider a zero-sum MG[1] $\mathcal{G}$ characterized by $(\mathcal{S}, \mathcal{A}, \mathcal{B}, P, r, \gamma)$, where $\mathcal{S}$ is the state space; $\mathcal{A}, \mathcal{B}$ are the action spaces of agents 1 and 2, respectively; $P : \mathcal{S} \times \mathcal{A} \times \mathcal{B} \to \Delta(\mathcal{S})$ denotes the transition probability of states; $r : \mathcal{S} \times \mathcal{A} \times \mathcal{B} \to [0, 1]$ denotes the reward function[2] of agent 1 (thus $-r$ is the bounded reward function of agent 2); and $\gamma \in [0, 1)$ is the discount factor. The goal of agent 1 (agent 2) is to maximize (minimize) the long-term accumulative discounted reward (a.k.a. return). In MARL, neither the transition nor the reward function is known to the agents.

Specifically, at each time $t$, agent 1 (agent 2) has a stationary (not necessarily deterministic) policy $\mu : \mathcal{S} \to \Delta(\mathcal{A})$ ($\nu : \mathcal{S} \to \Delta(\mathcal{B})$), where $\Delta(\mathcal{X})$ denotes the space of all probability measures over $\mathcal{X}$, so that $a_t \sim \mu(\cdot \mid s_t)$ ($b_t \sim \nu(\cdot \mid s_t)$). The state makes a transition from $s_t$ to $s_{t+1}$ following the probability distribution $P(\cdot \mid s_t, a_t, b_t)$, given $(a_t, b_t)$. As in the MDP model, one can define the *state-value function* under a pair of joint policies $(\mu, \nu)$ as

$$V^{\mu,\nu}(s) := \mathbb{E}_{a_t \sim \mu(\cdot \mid s_t), b_t \sim \nu(\cdot \mid s_t)}\left[ \sum_{t \geq 0} \gamma^t r(s_t, a_t, b_t) \,\middle|\, s_0 = s \right].$$

Note that $V^{\mu,\nu}(s) \in [0, 1/(1-\gamma)]$ for any $s \in \mathcal{S}$ as $r \in [0, 1]$, and the expectation is taken over the random trajectory produced by the joint policy $(\mu, \nu)$. Also, the *state-action/Q-value function* under $(\mu, \nu)$ is defined by

$$Q^{\mu,\nu}(s, a, b) := \mathbb{E}_{a_t \sim \mu(\cdot \mid s_t), b_t \sim \nu(\cdot \mid s_t)}\left[ \sum_{t \geq 0} \gamma^t r(s_t, a_t, b_t) \,\middle|\, s_0 = s, a_0 = a, b_0 = b \right].$$

The solution concept usually considered is the (approximate) *Nash equilibrium*, as defined below.

**Definition 2.1** (($\epsilon$-) Nash Equilibrium). For a zero-sum MG $(\mathcal{S}, \mathcal{A}, \mathcal{B}, P, r, \gamma)$, a *Nash equilibrium policy* pair $(\mu^*, \nu^*)$ satisfies the following pair of inequalities[3] for any $s \in \mathcal{S}$, $\mu \in \Delta(\mathcal{A})^{|\mathcal{S}|}$, and $\nu \in \Delta(\mathcal{B})^{|\mathcal{S}|}$

$$V^{\mu,\nu^*}(s) \leq V^{\mu^*,\nu^*}(s) \leq V^{\mu^*,\nu}(s). \tag{2.1}$$

If (2.1) holds with some $\epsilon > 0$ relaxation, i.e., for some policy $(\mu', \nu')$, such that

$$V^{\mu,\nu'}(s) - \epsilon \leq V^{\mu',\nu'}(s) \leq V^{\mu',\nu}(s) + \epsilon, \tag{2.2}$$

then $(\mu', \nu')$ is an $\epsilon$-*Nash equilibrium policy* pair.

By [20, 21], there exists a Nash equilibrium policy pair $(\mu^*, \nu^*) \in \Delta(\mathcal{A})^{|\mathcal{S}|} \times \Delta(\mathcal{B})^{|\mathcal{S}|}$ for two-player discounted zero-sum MGs. The state-value $V^* := V^{\mu^*,\nu^*}$ is referred to as the *value of the game*. The corresponding Q-value function is denoted by $Q^*$. The objective of the two agents is to find the NE policy of the MG, namely, to solve the saddle-point problem

$$\max_{\mu} \min_{\nu} \quad V^{\mu,\nu}(s), \tag{2.3}$$

for every $s \in \mathcal{S}$, where the order of $\max$ and $\min$ can be interchanged [46, 20]. For notational convenience, for any policy $(\mu, \nu)$, we define

$$V^{\mu, *} = \min_{\nu} V^{\mu, \nu}, \qquad\qquad V^{*, \nu} = \max_{\mu} V^{\mu, \nu}, \qquad\qquad (2.4)$$

and denote the corresponding optimizers to be $\nu(\mu)$ and $\mu(\nu)$, respectively. We refer to these values and optimizers as the *best-response* values and policies, given $\mu$ and $\nu$, respectively.

**Model-Based MARL**  As a standard setting, suppose that we have access to a *generative model/sampler*, which can provide us with samples $s' \sim P(\cdot \,|\, s, a, b)$ for any $(s, a, b)$. The model-based MARL algorithm simply calls the sampler $N$ times at each state-joint-action pair $(s, a, b)$, and estimates the actual game model $\mathcal{G}$ using data, by constructing an empirical model $\widehat{\mathcal{G}}$. Let $\widehat{P}$ denote the transition probability of the empirical model. Then $\widehat{P}$ is estimated by

$$\widehat{P}(s' \,|\, s, a, b) = \frac{\text{count}(s', s, a, b)}{N},$$

where $\text{count}(s', s, a, b)$ is the number of times the state-action pair $(s, a, b)$ forces a transition to state $s'$. Except for the transition model, all other elements in the empirical MG $\widehat{\mathcal{G}}$ are identical to those in $\mathcal{G}$.[4] We then use $\widehat{V}^{\mu, \nu}$, $\widehat{V}^{\mu, *}$, $\widehat{V}^{*, \nu}$, and $\widehat{V}^*$ to denote the value under $(\mu, \nu)$, the best-response value under $\mu$ and $\nu$, and the NE value, under the empirical game model $\widehat{\mathcal{G}}$, respectively. A similar convention is also used for Q-functions.

**Planning Oracle**  With a good empirical model $\widehat{\mathcal{G}}$ at hand, we further assume, as in [23] for single-agent RL, that we have an efficient planning oracle, which takes $\widehat{\mathcal{G}}$ as input, and outputs a policy pair $(\widehat{\mu}, \widehat{\nu})$. This oracle decouples the statistical and computational aspects of the estimate model $\widehat{\mathcal{G}}$. The output policy pair, referred to as being *near-equilibrium*, is assumed to satisfy certain $\epsilon_{opt}$-order of equilibrium, in terms of value functions, and we would like to evaluate the performance of $\widehat{\mu}, \widehat{\nu}$ on the original MG $\mathcal{G}$. Common planning algorithms include value iteration [20] and (generalized) policy iteration [21, 22], which are efficient in finding the ($\epsilon$-)NE of $\widehat{\mathcal{G}}$. In addition, it is not hard to have an oracle that is smooth in generating policies, i.e., the change of the approximate NE policies can be bounded by the changes of the NE value. See our Definition 3.4 later for a formal statement.

## 3  Main Results

We now introduce the main results of this paper. We first establish a lower bound on both approximating the NE value function, and learning the $\epsilon$-NE policy pair.

**Lemma 3.1** (Lower Bound)**.**  Let $\mathcal{G}$ be an unknown zero-sum MG. Then, there exist $\epsilon_0, \delta_0 > 0$, such that for all $\epsilon \in (0, \epsilon_0)$, $\delta \in (0, \delta_0)$, the sample complexity of learning an $\epsilon$-NE policy pair, or an $\epsilon$-approximate NE value, i.e., finding a $\widehat{Q}$ such that $\|\widehat{Q} - Q^*\|_\infty \le \epsilon$ for $\mathcal{G}$, with a generative model with probability at least $1 - \delta$, is

$$\Omega \left( \frac{|\mathcal{S}|(|\mathcal{A}| + |\mathcal{B}|)}{(1 - \gamma)^3 \epsilon^2} \log \left( \frac{1}{\delta} \right) \right).$$

The proof of Lemma 3.1, adapted from the proof of lower bound for MDPs [26, 47], is provided in §B. In the absence of the opponent, the lower bound reduces to that of a single-agent setting. As we will show momentarily, our sample complexity is nearly tight in $1 - \gamma$ and $|\mathcal{S}|$, while has a gap in $|\mathcal{A}|, |\mathcal{B}|$ dependence. We thus conjecture that the lower bound might still be improvable in $|\mathcal{A}|, |\mathcal{B}|$, but also highlight the challenges in generalizing the lower-bound proof for MDPs [26, 47] in §B. In the extended version of the present work (after the conference submission) [48], we have separated some *reward-agnostic* setting from the *reward-aware* setting, where in the former, the reward information is only revealed to the agent in the planning phase (not in the sampling phase). Indeed, this model-based approach can inherently handle both cases, as reward is not used in estimating $\widehat{P}$. In this more challenging reward-agnostic setting, we make the lower bound of $\widetilde{\Omega}\big(|\mathcal{S}||\mathcal{A}||\mathcal{B}|(1 - \gamma)^{-3}\epsilon^{-2}\big)$ possible. This separation seems unique in the multi-agent setting. See [48] for more details.

## 3.1 Near-Optimality in Finding $\epsilon$-Approximate NE Value

We now establish the near-minimax optimal sample complexities of model-based MARL. We start by showing the sample complexity to achieve an $\epsilon$-approximate NE value.

**Theorem 3.2** (Finding $\epsilon$-approximate NE Value). Suppose that the policy pair $(\widehat{\mu}, \widehat{\nu})$ is obtained from the **Planning Oracle** using the empirical model $\widehat{\mathcal{G}}$, which satisfies

$$\|\widehat{V}^{\widehat{\mu},\widehat{\nu}} - \widehat{V}^*\|_\infty \leq \epsilon_{opt}.$$

Then, for any $\delta \in (0,1]$ and $\epsilon \in (0, 1/(1-\gamma)^{1/2}]$, if

$$N \geq \frac{c\gamma \log\left[c|\mathcal{S}||\mathcal{A}||\mathcal{B}|(1-\gamma)^{-2}\delta^{-1}\right]}{(1-\gamma)^3\epsilon^2}$$

for some absolute constant $c$, it holds that with probability at least $1 - \delta$,

$$\left\|Q^{\widehat{\mu},\widehat{\nu}} - Q^*\right\|_\infty \leq \frac{2\epsilon}{3} + \frac{5\gamma\epsilon_{opt}}{1-\gamma}, \qquad \left\|\widehat{Q}^{\widehat{\mu},\widehat{\nu}} - Q^*\right\|_\infty \leq \epsilon + \frac{9\gamma\epsilon_{opt}}{1-\gamma}.$$

Theorem 3.2 shows that if the planning error $\epsilon_{opt}$ is made small, e.g., with the order of $\mathcal{O}((1-\gamma)\epsilon)$, then the Nash equilibrium Q-value can be estimated with a sample complexity of $\widetilde{\mathcal{O}}(|\mathcal{S}||\mathcal{A}||\mathcal{B}|(1-\gamma)^{-3}\epsilon^{-2})$, as $N$ queries are made for each $(s, a, b)$ pair. This planning error can be achieved by performing any efficient black-box optimization technique over the empirical model $\widehat{\mathcal{G}}$. Examples of such oracles include value iteration [20] and (generalized) policy iteration [21, 22]. Moreover, note that, in contrast to the single-agent setting, where only a $\max$ operator is used, a $\min\max$ (or $\max\min$) operator is used in these algorithms, which involves solving a *matrix game* at each state. This can be solved as a linear program [49], with at best polynomial runtime complexity [50, 51]. This in total leads to an efficient polynomial runtime complexity algorithm.

As per Lemma 3.1, our $\widetilde{\mathcal{O}}(|\mathcal{S}||\mathcal{A}||\mathcal{B}|(1-\gamma)^{-3}\epsilon^{-2})$ complexity is near-minimax optimal, in that it is near-tight in the dependence of $1-\gamma$ and $|\mathcal{S}|$, and sublinear in the model-size (which is $|\mathcal{S}|^2|\mathcal{A}||\mathcal{B}|$). The optimality in $1-\gamma$ and $|\mathcal{S}|$ dependence is significant, as in practice the discount factor can be very close to 1, and the large state-space size $|\mathcal{S}|$ is usually the bottleneck for algorithm-design. Moreover, if the action-space size of one agent dominates the other's (e.g., $|\mathcal{A}| \gg |\mathcal{B}|$), then our result is also optimal in the $|\mathcal{A}|, |\mathcal{B}|$ dependence. Unfortunately, without further assumption on the MG, e.g., being turn-based, the model-based algorithm can hardly avoid the $\mathcal{O}(|\mathcal{S}||\mathcal{A}||\mathcal{B}|)$ dependence, as it is required to estimate each $\widehat{P}(\cdot \,|\, s, a, b)$ accurately to perform the planning. Instead, as we discussed in §B, we suspect that the $\Omega(|\mathcal{A}| + |\mathcal{B}|)$ lower bound might be achievable by model-free MARL algorithms. Our result also matches the only known near-optimal sample complexity in zero-sum (but turn-based) MGs [10], which was achieved by a model-free Q-learning-based algorithm, with a $\mathcal{O}(|\mathcal{A}| + |\mathcal{B}|)$ dependence on $|\mathcal{A}|, |\mathcal{B}|$, as the transition $P$ is only controlled by one action in $\mathcal{A} \cup \mathcal{B}$.

However, this near-optimal result does not necessarily lead to near-optimal sample complexity for obtaining the $\epsilon$-NE *policies*. We first use a direct translation to obtain such an $\epsilon$-NE policy pair based on Theorem 3.2, for *any* **Planning Oracle**.

**Corollary 3.3** (Finding $\epsilon$-NE Policy). Let $(\widehat{\mu}, \widehat{\nu})$ and $N$ satisfy the conditions in Theorem 3.2. Let

$$\widetilde{\epsilon} := \frac{2}{1-\gamma} \cdot \left(\epsilon + \frac{9\gamma\epsilon_{opt}}{1-\gamma}\right),$$

and $(\widetilde{\mu}, \widetilde{\nu})$ be the one-step Nash equilibrium of $\widehat{Q}^{\widehat{\mu},\widehat{\nu}}$, namely, for any $s \in \mathcal{S}$

$$\left(\widetilde{\mu}(\cdot \,|\, s), \widetilde{\nu}(\cdot \,|\, s)\right) \in \operatorname*{argmax}_{u \in \Delta(\mathcal{A})} \min_{\vartheta \in \Delta(\mathcal{B})} \mathbb{E}_{a \sim u, b \sim \vartheta}\left[\widehat{Q}^{\widehat{\mu},\widehat{\nu}}(s, a, b)\right].$$

Then, with probability at least $1 - \delta$,

$$V^{*,\widetilde{\nu}} - 2\widetilde{\epsilon} \leq V^{\widetilde{\mu},\widetilde{\nu}} \leq V^{\widetilde{\mu},*} + 2\widetilde{\epsilon}, \tag{3.1}$$

namely, $(\widetilde{\mu}, \widetilde{\nu})$ constitutes a $2\widetilde{\epsilon}$-Nash equilibrium policy pair.

Corollary 3.3 is equivalently to saying that the sample complexity of achieving an $\epsilon$-NE policy pair is $\widetilde{\mathcal{O}}((1-\gamma)^{-5}\epsilon^{-2})$. This is worse than the model-based single-agent setting [23], and also worse than both the model-free single-agent [27] and turn-based two-agent [10] settings, where $\widetilde{\mathcal{O}}((1-\gamma)^{-3}\epsilon^{-2})$ can be achieved for learning the optimal policy. This also has a gap from the lower bound given in Lemma 3.1. Note that the above sample complexity still matches that of the Empirical QVI in [26] if $\epsilon \in (0, 1]$ for single-agent RL, but with a larger choice of $\epsilon$ of $(0, (1-\gamma)^{-1/2}]$. We also note that the Markov game setting is more challenging than MDPs, and using a general **Planning Oracle**, it is not clear so far if the lower bound given in Lemma 3.1 can be matched. In contrast, we show next that a stable **Planning Oracle** can indeed match the lower bound.

## 3.2 Near-Optimality in Finding $\epsilon$-NE Policy

Admittedly, the results in Corollary 3.3 do not fully exploit the *model-based* approach, since it finds the NE policy according to the Q-value estimate $\widehat{Q}^{\widehat{\mu},\widehat{\nu}}$, instead of using the output policy pair $(\widehat{\mu}, \widehat{\nu})$ directly. This loses a factor of $1 - \gamma$. To improve the sample complexity of obtaining the NE policies, we first introduce the following definition of a smooth **Planning Oracle**.

**Definition 3.4** (Smooth **Planning Oracle**). A smooth **Planning Oracle** generates policies that are smooth with respect to the NE Q-values of the empirical model. Specifically, for two empirical models $\widehat{\mathcal{G}}_1$ and $\widehat{\mathcal{G}}_2$, the generated near-equilibrium policy pair $(\widehat{\mu}_1, \widehat{\nu}_1)$ and $(\widehat{\mu}_2, \widehat{\nu}_2)$ satisfy that for each $s \in \mathcal{S}$, $\|\widehat{\mu}_1(\cdot\,|\,s) - \widehat{\mu}_2(\cdot\,|\,s)\|_{TV} \leq C \cdot \|\widehat{Q}_1^* - \widehat{Q}_2^*\|_\infty$ and $\|\widehat{\nu}_1(\cdot\,|\,s) - \widehat{\nu}_2(\cdot\,|\,s)\|_{TV} \leq C \cdot \|\widehat{Q}_1^* - \widehat{Q}_2^*\|_\infty$ for some $C > 0$, where $\widehat{Q}_i^*$ is the NE Q-value of $\widehat{\mathcal{G}}_i$ for $i = 1, 2$, and $\|\cdot\|_{TV}$ is the total variation distance.

Such a smooth **Planning Oracle** can be readily obtained in several ways. For example, one simple (but possibly computationally expensive) approach is to output the average over the entire policy space, using a *softmax* randomization over best-response values induced by $\widehat{Q}^*$. Specifically, for agent 1, the output $\widehat{\mu}$ is given by

$$\widehat{\mu}(\cdot\,|\,s) = \int_{\Delta(\mathcal{A})} \frac{\exp\big(\min_{\vartheta \in \Delta(\mathcal{B})} \mathbb{E}_{a \sim u, b \sim \vartheta}\big[\widehat{Q}^*(s, a, b)\big]/\tau\big)}{\int_{\Delta(\mathcal{A})} \exp\big(\min_{\vartheta \in \Delta(\mathcal{B})} \mathbb{E}_{a \sim u', b \sim \vartheta}\big[\widehat{Q}^*(s, a, b)\big]/\tau\big)du'} \cdot u\,du,$$

where $\tau > 0$ is some temperature constant. The output of $\widehat{\nu}$ is analogous. With a small enough $\tau$, $\widehat{\mu}$ approximates the exact solution to $\operatorname{argmax}_{u \in \Delta(\mathcal{A})} \min_{\vartheta \in \Delta(\mathcal{B})} \mathbb{E}_{a \sim u, b \sim \vartheta}[\widehat{Q}^*(s, a, b)]$, the NE policy given $\widehat{Q}^*$. Moreover, notice that $\widehat{\mu}$ satisfies the smoothness condition in Definition 3.4. This is because for each $u \in \Delta(\mathcal{A})$ in the integral: i) the softmax function is Lipschitz continuous with respect to the input $\min_{\vartheta \in \Delta(\mathcal{B})} \mathbb{E}_{a \sim u, b \sim \vartheta}\big[\widehat{Q}^*(s, a, b)\big]/\tau$ [52]; ii) the best-response value $\min_{\vartheta \in \Delta(\mathcal{B})} \mathbb{E}_{a \sim u, b \sim \vartheta}\big[\widehat{Q}^*(s, a, b)\big]$ is smooth with respect to $\widehat{Q}^*$. Thus, such an oracle is an instance of a smooth **Planning Oracle**.

Another more tractable way to obtain $(\widehat{\mu}, \widehat{\nu})$ is by solving a *regularized* matrix game induced by $\widehat{Q}^*$. Specifically, one solves

$$\big(\widehat{\mu}(\cdot\,|\,s), \widehat{\nu}(\cdot\,|\,s)\big) = \operatorname*{argmax}_{u \in \Delta(\mathcal{A})} \min_{\vartheta \in \Delta(\mathcal{B})} \mathbb{E}_{a \sim u, b \sim \vartheta}\big[\widehat{Q}^*(s, a, b)\big] - \tau_1 \Omega_1(u) + \tau_2 \Omega_2(\vartheta), \qquad (3.2)$$

for each state $s \in \mathcal{S}$, where $\Omega_i$ is the regularizer for agent $i$'s policy, usually a strongly convex function, $\tau_i > 0$ are the temperature parameters. Such a strongly-convex-strongly-concave saddle point problem admits a unique solution, and can be solved efficiently [53, 54, 55]. This regularized objective has been widely used in both single-agent MDPs [56, 57, 58, 59], and learning in games [60, 61, 62], with the advantages of having both better exploration and better convergence properties.

With small enough $\tau_i$, the solution to (3.2) will be close to that of the unregularized one [59], up to some error captured by $\epsilon_{opt}$. More importantly, many commonly used regularizations, including negative entropy [56], Tsallis entropy [58] and Rényi entropy with certain parameters [61], naturally yield a smooth **Planning Oracle**; see Lemma C.1 in the appendix for a formal statement. Note that the smoothness property of the oracle does not affect the sample complexity of our model-based MARL algorithm.

Now we are ready to present another theorem, which gives the $\epsilon$-Nash equilibrium *policy pair* directly, with the near-minimax optimal sample complexity of $\widetilde{\mathcal{O}}(|\mathcal{S}||\mathcal{A}||\mathcal{B}|(1-\gamma)^{-3}\epsilon^{-2})$.

**Theorem 3.5** (Finding $\epsilon$-NE Policy with a Smooth **Planning Oracle**). Suppose that the policy pair $(\widehat{\mu}, \widehat{\nu})$ is obtained from a smooth **Planning Oracle** using the empirical model $\widehat{\mathcal{G}}$ (see Definition 3.4), which satisfies

$$\|\widehat{V}^{\widehat{\mu},*} - \widehat{V}^*\|_\infty \le \epsilon_{opt}, \qquad \|\widehat{V}^{*,\widehat{\nu}} - \widehat{V}^*\|_\infty \le \epsilon_{opt}.$$

Then, for any $\delta \in (0,1]$ and $\epsilon \in (0, 1/(1-\gamma)^{1/2}]$, if

$$N \ge \frac{c\gamma \log\left[c(C+1)|\mathcal{S}||\mathcal{A}||\mathcal{B}|(1-\gamma)^{-4}\delta^{-1}\right]}{(1-\gamma)^3\epsilon^2}$$

for some absolute constant $c$, then, letting $\widetilde{\epsilon} := \epsilon + 4\epsilon_{opt}/(1-\gamma)$, with probability at least $1-\delta$,

$$V^{*,\widehat{\nu}} - 2\widetilde{\epsilon} \le V^{\widehat{\mu},\widehat{\nu}} \le V^{\widehat{\mu},*} + 2\widetilde{\epsilon},$$

namely, $(\widehat{\mu}, \widehat{\nu})$ constitutes a $2\widetilde{\epsilon}$-Nash equilibrium policy pair.

Theorem 3.5 shows that the sample complexity of achieving an $\epsilon$-NE policy can be near-minimax optimal, if a smooth **Planning Oracle** is used. This also matches the only known near-optimal sample complexity in MGs in [10], with a turn-based setting and a model-free algorithm. Inherited from [23], this improves the second result in [26] that also has $\widetilde{\mathcal{O}}((1-\gamma)^{-3}\epsilon^{-2})$ in finding an $\epsilon$-optimal policy, by removing the dependence on $|\mathcal{S}|^{-1/2}$ and enlarging the choice of $\epsilon$ from $(0, (1-\gamma)^{-1/2}|\mathcal{S}|^{-1/2})$ to $(0, (1-\gamma)^{-1/2}]$, and removing a factor of $|\mathcal{S}|$ in the total sample complexity for any fixed $\epsilon$. Theorems 3.2 and 3.5 together for the first time justify that, this simple model-based MARL algorithm is indeed sample-efficient, in approximating both the Nash equilibrium values and policies.

## 4  Sketch of Proofs

Detailed proofs are provided in the appendix. For the lower bound in Lemma 3.1, the proof is given in §B, together with some insights and comments that may benefit the tackling of sample complexity questions in zero-sum MGs in general. For the upper bounds, we provide below a proof roadmap:

**Proof Roadmap.**   Our proof for the upper bounds mainly consists of the following steps:

1. **Helper lemmas and a crude bound.** We first establish several important lemmas, including the component-wise error bounds for the final Q-value errors, the variance error bound, and a crude error bound that directly uses Hoeffding's inequality. Some of the results are adapted from the single-agent setting to zero-sum MGs. See §A.1.

2. **Establishing an auxiliary Markov game.** To improve the crude bound, we build up an *absorbing Markov game*, in order to handle the statistical dependence between $\widehat{P}$ and some value function generated by $\widehat{P}$, which occurs as a product in the component-wise bound above. By carefully designing the auxiliary game, we establish a Bernstein-like concentration inequality, despite this dependency. See §A.2, and more precisely, Lemmas A.9 and A.10.

3. **Final bound for $\epsilon$-approximate NE value.** Lemma A.9 in Step **2** allows us to exploit the variance bound, see Lemma A.3, to obtain an $\widetilde{\mathcal{O}}(\sqrt{1/[(1-\gamma)^3]N})$ order bound on the Q-value error, leading to a $\widetilde{\mathcal{O}}((1-\gamma)^{-3}\epsilon^{-2})$ near-minimax optimal sample complexity for achieving the $\epsilon$-approximate NE value. See §A.3.

4. **Final bounds for $\epsilon$-NE policy.** Based on the final bound in Step **3**, we then establish a $\widetilde{\mathcal{O}}((1-\gamma)^{-5}\epsilon^{-2})$ sample complexity for obtaining an $\epsilon$-NE policy pair, by solving an additional matrix game over the output Q-value $\widehat{Q}^{\widehat{\mu},\widehat{\nu}}$. See §A.4. In addition, given a smooth **Planning Oracle**, by Lemma A.10 in Step **2**, and more careful self-bounding techniques, we establish a $\widetilde{\mathcal{O}}((1-\gamma)^{-3}\epsilon^{-2})$ sample complexity for achieving such an $\epsilon$-NE policy pair, directly using the output policies $(\widehat{\mu}, \widehat{\nu})$. See §A.5.

The key in Step **1** is to establish the component-wise bounds for the error between the approximate and the actual NE value functions, which will finally give an $\epsilon$-approximate NE value/$\epsilon$-NE policy complexity. We consider here finding the $\epsilon$-NE policy as an example, and note that the inequalities

needed for finding the $\epsilon$-approximate NE value are similar, but without using the smoothness of the **Planning Oracle**. For agent 1, by the component-wise bounds (see Lemma A.1),

$$V^{\mu,*} \geq V^* - \|Q^{\mu,*} - \widehat{Q}^{\mu,*}\|_\infty - \|\widehat{V}^{\mu,*} - \widehat{V}^*\|_\infty - \|\widehat{Q}^{\mu^*,*} - Q^*\|_\infty. \tag{4.1}$$

Then, it suffices to quantify the bounds for $\|Q^{\mu,*} - \widehat{Q}^{\mu,*}\|_\infty$ and $\|\widehat{Q}^{\mu^*,*} - Q^*\|_\infty$, as the middle term in (4.1) is just the optimization error $\epsilon_{opt}$ from the **Planning Oracle**, which can be made very small. To this end, the following bound is important

$$\gamma(I - \gamma\widehat{P}^{\widehat{\mu},\nu(\widehat{\mu})})^{-1}(P - \widehat{P})V^{\widehat{\mu},*} \leq Q^{\widehat{\mu},*} - \widehat{Q}^{\widehat{\mu},*} \leq \gamma(I - \gamma P^{\widehat{\mu},\widehat{\nu(\widehat{\mu})}})^{-1}(P - \widehat{P})\widehat{V}^{\widehat{\mu},*}, \tag{4.2}$$

where $\widehat{\mu}$ is the output of the **Planning Oracle** using $\widehat{P}$, $\nu(\widehat{\mu})$ and $\widehat{\nu(\widehat{\mu})}$ denote the best-response policy of $\widehat{\mu}$ under the empirical and true models, respectively. The last important result in Step **1** is the following variance bound

$$\left\|(I - \gamma P^{\mu,\nu})^{-1}\sqrt{\mathrm{Var}_P\left(V_\mathcal{G}^{\mu,\nu}\right)}\right\|_\infty \leq \sqrt{\frac{2}{(1-\gamma)^3}}, \tag{4.3}$$

which will eventually give us $(1-\gamma)^{-3}$ dependence (the key for minimax optimality), if the terms $(P - \widehat{P})\widehat{V}^{\widehat{\mu},*}$ and $(P - \widehat{P})V^{\widehat{\mu},*}$ in (4.2) can be related to the variances of $\widehat{V}^{\widehat{\mu},*}$ and $V^{\widehat{\mu},*}$ in (4.3).

Nonetheless, the critical issue here is that $(P - \widehat{P})$ is statistically dependent on $\widehat{V}^{\widehat{\mu},*}$ and $V^{\widehat{\mu},*}$, preventing the use of Bernstein-like concentration inequality to connect the two. Step **2** is devoted to addressing this. We introduce a new Markov game $\mathcal{G}_{s,u}$ as follows (with $s \in \mathcal{S}$ and $u \in \mathbb{R}$ a constant): $\mathcal{G}_{s,u}$ is identical to $\mathcal{G}$, except that $P_{\mathcal{G}_{s,u}}(s \mid s, a, b) = 1$ for all $(a, b) \in \mathcal{A} \times \mathcal{B}$, namely, state $s$ is an *absorbing* state; and the instantaneous reward at $s$ is always $(1 - \gamma)u$. The rest of the reward function and the transition model of $\mathcal{G}_{s,u}$ are the same as those of $\mathcal{G}$. For simplicity, we use $X_{s,u}^{\mu,\nu}$ to denote $X_{\mathcal{G}_{s,u}}^{\mu,\nu}$, where $X$ can be either the value functions $Q$ and $V$, or the reward function $r$, under the model $\mathcal{G}_{s,u}$. Similarly, an absorbing game $\widehat{\mathcal{G}}_{s,u}$ is also established for the empirical model $\widehat{\mathcal{G}}$. Now the values/policies obtained from the absorbing models (those quantities with $(s, u)$), specifically, $\widehat{V}_{s,u}^*$ and $V^{\widehat{\mu}_{s,u},*}$, become independent of $P - \widehat{P}$, enabling the use of Bernstein concentration and the variance bound (4.3).

Finally, we do need to connect the quantities in the absorbing models (with $(s, u)$) back to the true ones in the non-absorbing ones (without $(s, u)$). As in [63], one can show that the best-response/NE values are *robust* to such small perturbations of $u$. Hence, one can construct an $\epsilon$-net with respect to this parameter $u$, to control the error between $\widehat{V}_{s,u}^*$ and $\widehat{V}^*$, and thus obtain the following Bernstein-like concentration bound

$$\left|(P - \widehat{P})\widehat{V}^*\right| \leq \sqrt{\frac{2\log\left(16|\mathcal{S}||\mathcal{A}||\mathcal{B}|/[(1-\gamma)^2\delta]\right) \cdot \mathrm{Var}_P(\widehat{V}^*)}{N}} + \Delta'_{\delta,N} \tag{4.4}$$

with some small order term $\Delta'_{\delta,N}$. By noticing that $\widehat{V}^*$ above can then be replaced by $\widehat{V}^{\widehat{\mu},*} + \widehat{V}^* - \widehat{V}^{\widehat{\mu},*}$, while the error $\widehat{V}^* - \widehat{V}^{\widehat{\mu},*}$ can be controlled by $\epsilon_{opt}$, we resolve the bound for the right-hand side of (4.2). More subtly, the left-hand side involving $V^{\widehat{\mu},*}$ is *not* in general robust to the small perturbation of $u$. Indeed, the NE policy $\widehat{\mu}_{s,u}$ obtained from $\widehat{\mathcal{G}}_{s,u}$ may vary a lot from $\widehat{\mu}$, making the value $V^{\widehat{\mu},*}$ deviate from $V^{\widehat{\mu}_{s,u},*}$. Using a smooth **Planning Oracle**, such a deviation can be carefully controlled, too, yielding a similar bound as (4.4), with $\widehat{V}^*$ being replaced by $V^{\widehat{\mu},*}$. Steps **3** and **4** combine the building blocks (4.1)-(4.4) above, and obtain the final near-optimal bounds by careful self-bounding techniques. The result for finding $\epsilon$-NE policy in Corollary (3.3) is built upon Theorem 3.2, by additionally quantifying the error caused by one-step Nash equilibrium operation on $\widehat{Q}^{\widehat{\mu},\widehat{\nu}}$.

## 5    Conclusion & Discussion

In this paper, we have established the first near-minimax optimal sample complexity for model-based MARL, in terms of both achieving the Nash equilibrium value and policy, when a generative model is available. Our setting has been focused on the basic model in MARL — infinite-horizon discounted zero-sum Markov games [17], and our techniques have been motivated by the recent "absorbing MDP" idea stemming from [23]. Our results naturally open up the following interesting future directions:

- *Closing the gap.* As mentioned in §3, the $\Omega(|\mathcal{A}||\mathcal{B}|)$ sample complexity seems inevitable for model-based approaches in general, due to estimating $\widehat{P}(\cdot \mid s, a, b)$; while as discussed in §B, the $\Omega(|\mathcal{A}| + |\mathcal{B}|)$ lower bound might be hard to improve. As such, to close the gap, it is imperative to either develop a MARL algorithm, possibly model-free, that attains $\mathcal{O}(|\mathcal{A}| + |\mathcal{B}|)$, or develop new techniques to improve the lower bound to $\mathcal{O}(|\mathcal{A}||\mathcal{B}|)$, possibly with some coupling among matrix games at different states.

- *Near-optimal model-free algorithms.* Besides the turn-based setting in [10], the near-minimax optimal sample complexity of model-free algorithms for general simultaneous-move zero-sum MGs is still open. It would be interesting to compare the results and dependence on the parameters with our model-based ones.

- *Near-optimality in other settings.* It would also be interesting to explore the near-optimal sample complexity or regret in other MARL settings, e.g., when no generative model is available, or for general-sum Markov games. Some of these questions have also been raised and partially answered recently in [36]. It would also be interesting to go beyond the tabular case and consider cases with function approximation.

## Broader Impact

We believe that researchers of multi-agent reinforcement learning (MARL), especially those who are interested in the theoretical foundations of MARL, would benefit from this work. In particular, prior to this work, though intuitive and widely-used, the sample efficiency, specifically the *minimax optimality* of the sample complexity, of this model-based approach had not been established for MARL. This work justified the efficiency of this simple method for the first time in the MARL setting. We have also raised several important open questions on the sample complexity of MARL in zero-sum Markov games in general, which open up some future research directions toward rigorous theoretical understandings of MARL. In contrast to the rich literature on the theory of *model-free* MARL algorithms, the theory of model-based ones is relatively lacking. Our results have advocated the use of model-based MARL due to its sample efficiency, which would benefit MARL practitioners when choosing between the two types of algorithms in practice. As a theory-oriented work, we do not believe that our research will cause any ethical issue, or put anyone at any disadvantage.

## Acknowledgments and Disclosure of Funding

The research of K.Z. and T.B. was supported in part by the US Army Research Laboratory (ARL) Cooperative Agreement W911NF-17-2-0196, and in part by the Office of Naval Research (ONR) MURI Grant N00014-16-1-2710. The research of S.K. was supported by the funding from the ONR award N00014-18-1-2247, and NSF Awards CCF-1703574 and CCF-1740551.

## Footnotes

[1] We will hereafter refer to this model simply as a *MG*.

[2] Our results can be generalized to other ranges of reward function by a standard reduction, see e.g., [27].

[3] In game theory, this pair is commonly referred to as "saddle-point inequalities".

[4]Without loss of generality, the reward function is assumed to be known. As argued in [23], the complexity of estimating $r$ that contributes to the total complexity is only a lower order term.

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
