[Supplementary Material]

# Supplementary Materials for "Model-Based Multi-Agent RL in Zero-Sum Markov Games with Near-Optimal Sample Complexity"

## A  Proofs of the Main Results

We first introduce some additional notations for convenience.

**Notations.**  For a matrix $X \in \mathbb{R}^{m \times n}$, $X \geq c$ for some scalar $c \in \mathbb{R}$ means that each element of $X$ is no-less than $c$. For a vector $x$, we use $(x^2)$, $\sqrt{x}$, $|x|$ to denote the component-wise square, square-root, and absolute value of $x$. We use $P_{(s,a,b),s'}$ to denote the transition probability $P(s' \mid s, a, b)$, and $P_{s,a,b}$ to denote the vector $P(\cdot \mid s, a, b)$. We use $\|u - v\|_{TV}$ to denote the total variation distance between two probability distributions $u, v \in \Delta(\mathcal{S})$ over a finite space $\mathcal{S}$, which is defined as $\|u - v\|_{TV} := \frac{1}{2} \sum_{s \in \mathcal{S}} |u(s) - v(s)|$. We also use $P^{\mu,\nu}$ to denote the transition probability of state-action pairs induced by the policy pair $(\mu, \nu)$, which is defined as

$$P^{\mu,\nu}_{(s,a,b),(s',a',b')} = \mu(a' \mid s')\nu(b' \mid s')P(s' \mid s, a, b).$$

Hence, the Q-value function can be written as

$$Q^{\mu,\nu} = r + \gamma P^{\mu,\nu}Q^{\mu,\nu} = (I - \gamma P^{\mu,\nu})^{-1}r.$$

Also, for any $V \in \mathbb{R}^{|\mathcal{S}|}$, we define the vector $\mathrm{Var}_P(V) \in \mathbb{R}^{|\mathcal{S}| \times |\mathcal{A}| \times |\mathcal{B}|}$ as

$$\mathrm{Var}_P(V)(s,a,b) := \mathrm{Var}_{P(\cdot \mid s,a,b)}(V) = P(V)^2 - (PV)^2.$$

Then, we define $\Sigma^{\mu,\nu}_{\mathcal{G}}$ to be the variance of the discounted reward under the MG $\mathcal{G}$, i.e.,

$$\Sigma^{\mu,\nu}_{\mathcal{G}}(s,a,b) := \mathbb{E}\Big[\Big(\sum_{t=0}^{\infty} \gamma^t r(s_t, a_t, b_t) - Q^{\mu,\nu}_{\mathcal{G}}(s,a,b)\Big)^2 \,\Big|\, s_0 = s, a_0 = a, b_0 = b\Big].$$

It can be shown (see an almost identical formula for MDPs in [26, Lemma 6]) that $\Sigma^{\mu,\nu}_{\mathcal{G}}$ satisfies some *Bellman-type* equation for any policy pair $(\mu, \nu)$:

$$\Sigma^{\mu,\nu}_{\mathcal{G}} = \gamma^2 \, \mathrm{Var}_P(V^{\mu,\nu}_{\mathcal{G}}) + \gamma^2 P^{\mu,\nu}\Sigma^{\mu,\nu}_{\mathcal{G}}. \tag{A.1}$$

It can also be verified that $\|\Sigma^{\mu,\nu}_{\mathcal{G}}\|_\infty \leq \gamma^2/(1-\gamma)^2$ [26, 23]. Before proceeding further, we provide a roadmap of the proof.

**Proof Roadmap.**  Our proof mainly consists of the following steps:

1. **Helper lemmas and a crude bound.** We first establish several important lemmas, including the component-wise error bounds for the final Q-value errors, the variance error bound, and a crude error bound that directly uses Hoeffding's inequality. Some of the results are adapted from the single-agent setting to zero-sum MGs. See §A.1.

2. **Establishing an auxiliary Markov game.** To improve the crude bound, we build up an *absorbing Markov game*, in order to handle the statistical dependence between $\widehat{P}$ and some value function generated by $\widehat{P}$, which occurs as a product in the component-wise bound above. By carefully designing the auxiliary game, we establish a Bernstein-like concentration inequality, despite this dependency. See §A.2, and more precisely, Lemmas A.9 and A.10.

3. **Final bound for $\epsilon$-approximate NE value.** Lemma A.9 in Step **2** allows us to exploit the variance bound, see Lemma A.3, to obtain an $\widetilde{\mathcal{O}}(\sqrt{1/[(1-\gamma)^3]N})$ order bound on the Q-value error, leading to a $\widetilde{\mathcal{O}}((1-\gamma)^{-3}\epsilon^{-2})$ near-minimax optimal sample complexity for achieving the $\epsilon$-approximate NE value. See §A.3.

4. **Final bounds for $\epsilon$-NE policy.** Based on the final bound in Step **3**, we then establish a $\widetilde{\mathcal{O}}((1-\gamma)^{-5}\epsilon^{-2})$ sample complexity for obtaining an $\epsilon$-NE policy pair, by solving an additional matrix game over the output Q-value $\widehat{Q}^{\widehat{\mu},\widehat{\nu}}$. See §A.4. In addition, given a smooth **Planning Oracle**, by Lemma A.10 in Step **2**, and more careful self-bounding techniques, we establish a $\widetilde{\mathcal{O}}((1-\gamma)^{-3}\epsilon^{-2})$ sample complexity for achieving such an $\epsilon$-NE policy pair, directly using the output policies $(\widehat{\mu}, \widehat{\nu})$. See §A.5.

### A.1 Important Lemmas

We start with the component-wise error bounds.

**Lemma A.1** (Component-Wise Bounds). For any policy pair $(\mu, \nu)$, it follows that

$$Q^{\mu,\nu} - \widehat{Q}^{\mu,\nu} = \gamma(I - \gamma P^{\mu,\nu})^{-1}(P - \widehat{P})\widehat{V}^{\mu,\nu},$$

$$\gamma(I - \gamma P^{\mu,\nu(\mu)})^{-1}(P - \widehat{P})\widehat{V}^{\mu,\nu(\mu)} \leq Q^{\mu,*} - \widehat{Q}^{\mu,*} \leq \gamma(I - \gamma P^{\mu,\widehat{\nu(\mu)}})^{-1}(P - \widehat{P})\widehat{V}^{\mu,*},$$

$$\gamma(I - \gamma P^{\widehat{\mu(\nu)},\nu})^{-1}(P - \widehat{P})\widehat{V}^{*,\nu} \leq Q^{*,\nu} - \widehat{Q}^{*,\nu} \leq \gamma(I - \gamma \widehat{P}^{\mu(\nu),\nu})^{-1}(P - \widehat{P})V^{*,\nu},$$

where we recall that $\nu(\mu)$ and $\mu(\nu)$ denote the best-response policy given $\mu$ and $\nu$, respectively (see (2.4)). Moreover, we have

$$Q^{\mu,\nu} \geq Q^* - \|Q^{\mu,\nu} - \widehat{Q}^{\mu,\nu}\|_\infty - \|\widehat{Q}^{\mu,\nu} - \widehat{Q}^*\|_\infty - \|\widehat{Q}^{\mu^*,*} - Q^*\|_\infty \qquad (A.2)$$

$$Q^{\mu,\nu} \leq Q^* + \|Q^{\mu,\nu} - \widehat{Q}^{\mu,\nu}\|_\infty + \|\widehat{Q}^{\mu,\nu} - \widehat{Q}^*\|_\infty + \|\widehat{Q}^{*,\nu^*} - Q^*\|_\infty \qquad (A.3)$$

$$V^{\mu,*} \geq V^* - \|Q^{\mu,*} - \widehat{Q}^{\mu,*}\|_\infty - \|\widehat{V}^{\mu,*} - \widehat{V}^*\|_\infty - \|\widehat{Q}^{\mu^*,*} - Q^*\|_\infty \qquad (A.4)$$

$$V^{*,\nu} \leq V^* + \|Q^{*,\nu} - \widehat{Q}^{*,\nu}\|_\infty + \|\widehat{V}^{*,\nu} - \widehat{V}^*\|_\infty + \|\widehat{Q}^{*,\nu^*} - Q^*\|_\infty. \qquad (A.5)$$

*Proof.* First, note that

$$\begin{aligned}
Q^{\mu,\nu} - \widehat{Q}^{\mu,\nu} &= (I - \gamma P^{\mu,\nu})^{-1}r - (I - \gamma \widehat{P}^{\mu,\nu})^{-1}r \\
&= (I - \gamma P^{\mu,\nu})^{-1}[(I - \gamma \widehat{P}^{\mu,\nu}) - (I - \gamma P^{\mu,\nu})]\widehat{Q}^{\mu,\nu} \\
&= \gamma(I - \gamma P^{\mu,\nu})^{-1}(P^{\mu,\nu} - \widehat{P}^{\mu,\nu})\widehat{Q}^{\mu,\nu} = \gamma(I - \gamma P^{\mu,\nu})^{-1}(P - \widehat{P})\widehat{V}^{\mu,\nu},
\end{aligned}$$

proving the first equation. Also,

$$\begin{aligned}
Q^{\mu,*} - \widehat{Q}^{\mu,*} &\leq Q^{\mu,\widehat{\nu(\mu)}} - \widehat{Q}^{\mu,*} = Q^{\mu,\widehat{\nu(\mu)}} - \widehat{Q}^{\mu,\widehat{\nu(\mu)}} \\
&= \left(I - \gamma P^{\mu,\widehat{\nu(\mu)}}\right)^{-1}r - \left(I - \gamma \widehat{P}^{\mu,\widehat{\nu(\mu)}}\right)^{-1}r \\
&= \left(I - \gamma P^{\mu,\widehat{\nu(\mu)}}\right)^{-1}\left[(I - \gamma \widehat{P}^{\mu,\widehat{\nu(\mu)}}) - (I - \gamma P^{\mu,\widehat{\nu(\mu)}})\right]\widehat{Q}^{\mu,\widehat{\nu(\mu)}} \\
&= \gamma(I - \gamma P^{\mu,\widehat{\nu(\mu)}})^{-1}(P - \widehat{P})\widehat{V}^{\mu,\widehat{\nu(\mu)}},
\end{aligned}$$

where we recall that $\widehat{\nu(\mu)}(\cdot\,|\,s) \in \operatorname{argmin}\widehat{V}^{\mu,\nu}(s)$ for all $s \in \mathcal{S}$. By similar arguments, recalling that $\nu(\mu)(\cdot\,|\,s) \in \operatorname{argmin} V^{\mu,\nu}(s)$ for all $s$, we have

$$\begin{aligned}
Q^{\mu,*} - \widehat{Q}^{\mu,*} &\geq Q^{\mu,\nu(\mu)} - \widehat{Q}^{\mu,\nu(\mu)} = (I - \gamma P^{\mu,\nu(\mu)})^{-1}r - (I - \gamma \widehat{P}^{\mu,\nu(\mu)})^{-1}r \\
&= (I - \gamma \widehat{P}^{\mu,\nu(\mu)})^{-1}[(I - \gamma \widehat{P}^{\mu,\nu(\mu)}) - (I - \gamma P^{\mu,\nu(\mu)})]Q^{\mu,\nu(\mu)} \\
&= \gamma(I - \gamma \widehat{P}^{\mu,\nu(\mu)})^{-1}(P - \widehat{P})V^{\mu,*}.
\end{aligned}$$

Similar arguments yield the third inequality in the first argument.

For the second argument, we have

$$\begin{aligned}
Q^{\mu,\nu} - Q^* = Q^{\mu,\nu} - \widehat{Q}^* + \widehat{Q}^* - Q^* &\geq Q^{\mu,\nu} - \widehat{Q}^* + \widehat{Q}^{\mu^*,*} - Q^* \\
&\geq -\|Q^{\mu,\nu} - \widehat{Q}^*\|_\infty - \|\widehat{Q}^{\mu^*,*} - Q^*\|_\infty,
\end{aligned}$$

which, combined with triangle inequality, yields the first inequality. Similarly, we have

$$\begin{aligned}
Q^{\mu,\nu} - Q^* = Q^{\mu,\nu} - \widehat{Q}^* + \widehat{Q}^* - Q^* &\leq Q^{\mu,\nu} - \widehat{Q}^* + \widehat{Q}^{*,\nu^*} - Q^* \\
&\leq \|Q^{\mu,\nu} - \widehat{Q}^*\|_\infty + \|\widehat{Q}^{*,\nu^*} - Q^*\|_\infty,
\end{aligned}$$

Using triangle inequality proves the second inequality. For (A.4)-(A.5), we similarly have

$$\begin{aligned}
V^{\mu,*} - V^* = V^{\mu,*} - \widehat{V}^* + \widehat{V}^* - V^* &\geq V^{\mu,*} - \widehat{V}^* + \widehat{V}^{\mu^*,*} - V^* \\
&\geq -\|V^{\mu,*} - \widehat{V}^*\|_\infty - \|\widehat{V}^{\mu^*,*} - V^*\|_\infty, \qquad (A.6)
\end{aligned}$$

$$\begin{aligned}
V^{*,\nu} - V^* = V^{*,\nu} - \widehat{V}^* + \widehat{V}^* - V^* &\leq V^{*,\nu} - \widehat{V}^* + \widehat{V}^{*,\nu} - V^* \\
&\leq \|V^{*,\nu} - \widehat{V}^*\|_\infty + \|\widehat{V}^{*,\nu} - V^*\|_\infty. \qquad (A.7)
\end{aligned}$$

Notice that for any $\mu \in \Delta(\mathcal{A})^{|\mathcal{S}|}$ and $\nu \in \Delta(\mathcal{B})^{|\mathcal{S}|}$,

$$
\begin{aligned}
\|V^{\mu,*} &- \widehat{V}^{\mu,*}\|_\infty \\
&= \Big\| \min_{\vartheta \in \Delta(\mathcal{B})} \mathbb{E}_{a\sim\mu(\cdot\,|\,s),b\sim\vartheta}[Q^{\mu,*}(\cdot,a,b)] - \min_{\vartheta \in \Delta(\mathcal{B})} \mathbb{E}_{a\sim\mu(\cdot\,|\,s),b\sim\vartheta}[\widehat{Q}^{\mu,*}(\cdot,a,b)] \Big\|_\infty \\
&\leq \max_{\vartheta \in \Delta(\mathcal{B})} \big\| \mathbb{E}_{a\sim\mu(\cdot\,|\,s),b\sim\vartheta}[Q^{\mu,*}(\cdot,a,b)] - \mathbb{E}_{a\sim\mu(\cdot\,|\,s),b\sim\vartheta}[\widehat{Q}^{\mu,*}(\cdot,a,b)] \big\|_\infty \leq \|Q^{\mu,*} - \widehat{Q}^{\mu,*}\|_\infty
\end{aligned}
$$
(A.8)

$$
\begin{aligned}
\|V^{*,\nu} &- \widehat{V}^{*,\nu}\|_\infty \\
&= \Big\| \max_{u \in \Delta(\mathcal{A})} \mathbb{E}_{a\sim u,b\sim\nu(\cdot\,|\,s)}[Q^{*,\nu}(\cdot,a,b)] - \max_{u \in \Delta(\mathcal{A})} \mathbb{E}_{a\sim u,b\sim\nu(\cdot\,|\,s)}[\widehat{Q}^{*,\nu}(\cdot,a,b)] \Big\|_\infty \\
&\leq \max_{u \in \Delta(\mathcal{A})} \big\| \mathbb{E}_{a\sim u,b\sim\nu(\cdot\,|\,s)}[Q^{*,\nu}(\cdot,a,b)] - \mathbb{E}_{a\sim u,b\sim\nu(\cdot\,|\,s)}[\widehat{Q}^{*,\nu}(\cdot,a,b)] \big\|_\infty \leq \|Q^{*,\nu} - \widehat{Q}^{*,\nu}\|_\infty.
\end{aligned}
$$
(A.9)

Combining (A.6)-(A.7) and (A.8)-(A.9), together with triangle inequality, we arrive at (A.4)-(A.5), and complete the proof. $\qquad\square$

The errors in (A.2)-(A.3) are decomposed into three terms. The second term $\|\widehat{Q}^{\mu,\nu} - \widehat{Q}^*\|_\infty$ is the *optimization error* we obtained from the algorithm that solves the empirical game. This can be handled by the algorithm. We will thus focus on bounding the other two terms. To this end, we need the following lemma; see also Lemma 2 in [23].

**Lemma A.2.** For any policy pair $(\mu, \nu)$ and vector $v \in \mathbb{R}^{|\mathcal{S}|\times|\mathcal{A}|\times|\mathcal{B}|}$, $\|(I - \gamma P^{\mu,\nu})^{-1}v\|_\infty \leq \|v\|_\infty/(1-\gamma)$.

*Proof.* The proof is straightforward. Letting $w = (I - \gamma P^{\mu,\nu})^{-1}v$, we have $v = (I - \gamma P^{\mu,\nu})w$. Triangle inequality yields $\|v\|_\infty \geq \|w\|_\infty - \gamma\|P^{\mu,\nu}w\|_\infty \geq \|w\|_\infty - \gamma\|w\|_\infty$, which completes the proof. $\qquad\square$

Next we establish the Bellman property of a policy pair $(\mu, \nu)$'s variance and its accumulation. This has been observed for MDPs before in [64, 65, 66, 23]. We establish the counterpart for Markov games as follows.

**Lemma A.3.** For any policy pair $(\mu, \nu)$ and MG $\mathcal{G}$ with transition model $P$, we have

$$
\left\| (I - \gamma P^{\mu,\nu})^{-1} \sqrt{\mathrm{Var}_P\left(V_{\mathcal{G}}^{\mu,\nu}\right)} \right\|_\infty \leq \sqrt{\frac{2}{(1-\gamma)^3}}.
$$

*Proof.* The proof follows that of [23, Lemma 3]. For any positive vector $v$, by Jensen's inequality, we have

$$
\|(I - \gamma P^{\mu,\nu})^{-1}\sqrt{v}\|_\infty = \frac{1}{1-\gamma}\|(1-\gamma)(I - \gamma P^{\mu,\nu})^{-1}\sqrt{v}\|_\infty \leq \sqrt{\left\| \frac{1}{1-\gamma}(I - \gamma P^{\mu,\nu})^{-1}v \right\|_\infty}.
$$
(A.10)

Also, observe that

$$
\begin{aligned}
\|(I - \gamma P^{\mu,\nu})^{-1}v\|_\infty &= \|(I - \gamma P^{\mu,\nu})^{-1}(I - \gamma^2 P^{\mu,\nu})(I - \gamma^2 P^{\mu,\nu})^{-1}v\|_\infty \\
&= \big\|[(I - \gamma P^{\mu,\nu})^{-1}(1 - \gamma + \gamma - \gamma^2 P^{\mu,\nu})](I - \gamma^2 P^{\mu,\nu})^{-1}v\big\|_\infty \\
&= \big\|[(1-\gamma)(I - \gamma P^{\mu,\nu})^{-1} + \gamma I](I - \gamma^2 P^{\mu,\nu})^{-1}v\big\|_\infty \\
&\leq (1-\gamma)\big\|(I - \gamma P^{\mu,\nu})^{-1}(I - \gamma^2 P^{\mu,\nu})^{-1}v\big\|_\infty + \gamma\big\|(I - \gamma^2 P^{\mu,\nu})^{-1}v\big\|_\infty \\
&\leq \frac{1-\gamma}{1-\gamma}\big\|(I - \gamma^2 P^{\mu,\nu})^{-1}v\big\|_\infty + \gamma\big\|(I - \gamma^2 P^{\mu,\nu})^{-1}v\big\|_\infty \leq 2\big\|(I - \gamma^2 P^{\mu,\nu})^{-1}v\big\|_\infty. \quad \text{(A.11)}
\end{aligned}
$$

Combining (A.10) and (A.11) yields

$$
\|(I - \gamma P^{\mu,\nu})^{-1}\sqrt{v}\|_\infty \leq \sqrt{\left\| \frac{2}{1-\gamma}(I - \gamma^2 P^{\mu,\nu})^{-1}v \right\|_\infty}.
$$
(A.12)

In addition, by (A.1), we have $\Sigma_{\mathcal{G}}^{\mu,\nu} = \gamma^2(I - \gamma^2 P^{\mu,\nu})^{-1} \operatorname{Var}_P(V_{\mathcal{G}}^{\mu,\nu})$. Letting $v = \operatorname{Var}_P(V_{\mathcal{G}}^{\mu,\nu})$ in (A.12) and noticing that $\|\Sigma_{\mathcal{G}}^{\mu,\nu}\|_\infty \le \gamma^2/(1-\gamma)^2$ completes the proof. $\qquad\square$

Finally, if we just apply Hoeffding's inequality, we obtain the following concentration argument, upon which we will improve to obtain our final results.

**Lemma A.4.** Let $(\mu^*, \nu^*)$ be the Nash equilibrium policy pair under the actual model $\mathcal{G}$. Then, for any $\delta \in (0,1]$, with probability at least $1 - \delta$, we have

$$\|Q^* - \widehat{Q}^{\mu^*,\nu^*}\|_\infty \le \Delta_{\delta,N}, \qquad \|Q^* - \widehat{Q}^{\mu^*,*}\|_\infty \le \Delta_{\delta,N},$$
$$\|Q^* - \widehat{Q}^{*,\nu^*}\|_\infty \le \Delta_{\delta,N}, \qquad \|Q^* - \widehat{Q}^*\|_\infty \le \Delta_{\delta,N},$$

where

$$\Delta_{\delta,N} := \frac{\gamma}{(1-\gamma)^2} \sqrt{\frac{2\log(2|\mathcal{S}||\mathcal{A}||\mathcal{B}|/\delta)}{N}}.$$

*Proof.* First note that $V^*$ is fixed and independent of the randomness in $\widehat{P}$. Due to the boundedness of $V^*$ that $\|V^*\|_\infty \le (1-\gamma)^{-1}$, and the union of Hoeffding bounds over $\mathcal{S} \times \mathcal{A} \times \mathcal{B}$, we have that with probability at least $1 - \delta$

$$\|(\widehat{P} - P)V^*\|_\infty \le \frac{1}{1-\gamma} \cdot \sqrt{\frac{2\log(2|\mathcal{S}||\mathcal{A}||\mathcal{B}|/\delta)}{N}}. \tag{A.13}$$

On the other hand, let $\mathcal{T}_{\mu,\nu}$ be the Bellman operator under the true transition model $P$, using any joint policy $(\mu, \nu)$, i.e., for any $s \in \mathcal{S}$ and $(s,a,b) \in \mathcal{S} \times \mathcal{A} \times \mathcal{B}$, $V \in \mathbb{R}^{|\mathcal{S}|}$ and $Q \in \mathbb{R}^{|\mathcal{S}| \times |\mathcal{A}| \times |\mathcal{B}|}$:

$$\mathcal{T}_{\mu,\nu}(V)(s) = \mathbb{E}_{a \sim \mu(\cdot \mid s), b \sim \nu(\cdot \mid s)}\big[r(s,a,b) + \gamma \cdot P(\cdot \mid s,a,b)^\top V\big]$$
$$\mathcal{T}_{\mu,\nu}(Q)(s,a,b) = r(s,a,b) + \gamma \cdot \mathbb{E}_{s' \sim P(\cdot \mid s,a,b), a' \sim \mu(\cdot \mid s'), b' \sim \nu(\cdot \mid s')}\big[Q(s',a',b')\big].$$

Similarly, let $\widehat{\mathcal{T}}_{\mu,\nu}$ be the corresponding operator defined under the estimated transition $\widehat{P}$. Note that $\widehat{Q}^{\mu,\nu}$ and $Q^*$ are the fixed points of $\widehat{\mathcal{T}}_{\mu,\nu}$ and $\mathcal{T}_{\mu^*,\nu^*}$, respectively. We thus have

$$\begin{aligned}
\|Q^* - \widehat{Q}^{\mu,\nu}\|_\infty &= \|\mathcal{T}_{\mu^*,\nu^*} Q^* - \widehat{\mathcal{T}}_{\mu,\nu} \widehat{Q}^{\mu,\nu}\|_\infty \\
&\le \|\mathcal{T}_{\mu^*,\nu^*} Q^* - r - \gamma \widehat{P}^{\mu^*,\nu^*} Q^*\|_\infty + \|r + \gamma \widehat{P}^{\mu^*,\nu^*} Q^* - \widehat{\mathcal{T}}_{\mu,\nu} \widehat{Q}^{\mu,\nu}\|_\infty \\
&= \gamma \|P^{\mu^*,\nu^*} Q^* - \widehat{P}^{\mu^*,\nu^*} Q^*\|_\infty + \gamma \|\widehat{P}^{\mu^*,\nu^*} Q^* - \widehat{P}^{\mu,\nu} \widehat{Q}^{\mu,\nu}\|_\infty \\
&= \gamma \|PV^* - \widehat{P}V^*\|_\infty + \gamma \|\widehat{P}V^* - \widehat{P}\widehat{V}^{\mu,\nu}\|_\infty \le \gamma \|(P - \widehat{P})V^*\|_\infty + \gamma \|V^* - \widehat{V}^{\mu,\nu}\|_\infty.
\end{aligned} \tag{A.14}$$

To show the first argument, letting $\mu = \mu^*$ and $\nu = \nu^*$, we have

$$\begin{aligned}
\gamma \|V^* - \widehat{V}^{\mu^*,\nu^*}\|_\infty &\\
&= \gamma \big\|\mathbb{E}_{a \sim \mu^*(\cdot \mid s), b \sim \nu^*(\cdot \mid s)}[Q^*(\cdot,a,b)] - \mathbb{E}_{a \sim \mu^*(\cdot \mid s), b \sim \nu^*(\cdot \mid s)}[\widehat{Q}^{\mu^*,\nu^*}(\cdot,a,b)]\big\|_\infty \\
&\le \gamma \|Q^* - \widehat{Q}^{\mu^*,\nu^*}\|_\infty.
\end{aligned} \tag{A.15}$$

Using (A.15) to bound the last term in (A.14), and solving for $\|Q^* - \widehat{Q}^{\mu^*,\nu^*}\|_\infty$ from (A.14), we obtain the first argument.

For the second argument, letting $\mu = \mu^*$ and $\nu = \widehat{\nu(\mu^*)}$ (note that $\widehat{Q}^{\mu^*,*} = \widehat{Q}^{\mu^*,\widehat{\nu(\mu^*)}}$), we have

$$\begin{aligned}
\gamma \|V^* - \widehat{V}^{\mu^*,*}\|_\infty &\\
&= \gamma \big\| \min_{\vartheta \in \Delta(\mathcal{B})} \mathbb{E}_{a \sim \mu^*(\cdot \mid s), b \sim \vartheta}[Q^*(\cdot,a,b)] - \min_{\vartheta \in \Delta(\mathcal{B})} \mathbb{E}_{a \sim \mu^*(\cdot \mid s), b \sim \vartheta}[\widehat{Q}^{\mu^*,*}(\cdot,a,b)]\big\|_\infty \\
&\le \gamma \max_{\vartheta \in \Delta(\mathcal{B})} \big\| \mathbb{E}_{a \sim \mu^*(\cdot \mid s), b \sim \vartheta}[Q^*(\cdot,a,b)] - \mathbb{E}_{a \sim \mu^*(\cdot \mid s), b \sim \vartheta}[\widehat{Q}^{\mu^*,*}(\cdot,a,b)]\big\|_\infty \\
&\le \gamma \|Q^* - \widehat{Q}^{\mu^*,*}\|_\infty,
\end{aligned} \tag{A.16}$$

where the first inequality is due to the non-expansiveness of the $\min$ operator. Using (A.16) to bound the last term in (A.14), and solving for $\|Q^* - \widehat{Q}^{\widehat{\mu}^*,*}\|_\infty$ from (A.14), we obtain the second argument. Similarly, we can obtain the third argument.

For the fourth argument, letting $\mu = \widehat{\mu}^*$ and $\nu = \widehat{\nu}^*$, the NE policy under $\widehat{P}$ (note that $\widehat{Q}^{\widehat{\mu}^*,\widehat{\nu}^*} = \widehat{Q}^*$), we have

$$\gamma\|V^* - \widehat{V}^*\|_\infty$$
$$= \gamma\Big\|\max_{u \in \Delta(\mathcal{A})}\min_{\vartheta \in \Delta(\mathcal{B})}\mathbb{E}_{a \sim u, b \sim \vartheta}[Q^*(\cdot, a, b)] - \max_{u \in \Delta(\mathcal{A})}\min_{\vartheta \in \Delta(\mathcal{B})}\mathbb{E}_{a \sim u, b \sim \vartheta}[\widehat{Q}^*(\cdot, a, b)]\Big\|_\infty$$
$$\leq \gamma\max_{u \in \Delta(\mathcal{A})}\Big\|\min_{\vartheta \in \Delta(\mathcal{B})}\mathbb{E}_{a \sim u, b \sim \vartheta}[Q^*(\cdot, a, b)] - \min_{\vartheta \in \Delta(\mathcal{B})}\mathbb{E}_{a \sim u, b \sim \vartheta}[\widehat{Q}^*(\cdot, a, b)]\Big\|_\infty$$
$$\leq \gamma\|Q^* - \widehat{Q}^*\|_\infty,$$

where the inequalities are due to the non-expansivenesses of both the $\max$ and the $\min$ operators. This, combined with (A.14), completes the proof. $\qquad\square$

## A.2 An Auxiliary Markov Game

Motivated by the *absorbing MDP* technique in [23], we propose to introduce an *absorbing Markov game*, in order to handle the interdependence between $\widehat{P}$ and $\widehat{V}^{\mu,\nu}$, for any $\mu, \nu$ (which may also depend on $\widehat{P}$), which will show up frequently in the analysis.

We now define a new Markov game $\mathcal{G}_{s,u}$ as follows (with $s \in \mathcal{S}$ and $u \in \mathbb{R}$ a constant): $\mathcal{G}_{s,u}$ is identical to $\mathcal{G}$, except that $P_{\mathcal{G}_{s,u}}(s \,|\, s, a, b) = 1$ for all $(a, b) \in \mathcal{A} \times \mathcal{B}$, namely, state $s$ is an *absorbing* state; and the instantaneous reward at $s$ is always $(1 - \gamma)u$. The rest of the reward function and the transition model of $\mathcal{G}_{s,u}$ are the same as those of $\mathcal{G}$. For notational simplicity, we now use $X_{s,u}^{\mu,\nu}$ to denote $X_{\mathcal{G}_{s,u}}^{\mu,\nu}$, where $X$ can be either the value functions $Q$ and $V$, or the reward function $r$, under the model $\mathcal{G}_{s,u}$. Obviously, for any policy pair $(\mu, \nu)$, $V_{s,u}^{\mu,\nu}(s) = u$ for the absorbing state $s$.

In addition, we define $U_s$ for some state $s$ to choose $u$ from, which is a set of evenly spaced elements in the interval $[V^*(s) - \Delta, V^*(s) + \Delta]$ for some $\Delta > 0$, i.e., $U_s \subset [V^*(s) - \Delta, V^*(s) + \Delta]$. An appropriately chosen size of $|U_s|$ will be the key in the proof. We also use $\widehat{P}_{\mathcal{G}_{s,u}}$ to denote the transition model of the absorbing MG for the empirical MG $\widehat{\mathcal{G}}$, denoted by $\widehat{\mathcal{G}}_{s,u}$. Specifically, at all non-absorbing states, $\widehat{P}_{\mathcal{G}_{s,u}}$ is identical to $\widehat{P}$; while at the absorbing state, $\widehat{P}_{\mathcal{G}_{s,u}}(s \,|\, s, a, b) = 1$ for any $(a, b) \in \mathcal{A} \times \mathcal{B}$. The corresponding value functions are for short denoted by $\widehat{V}_{s,u}^{\mu,\nu}$ and $\widehat{Q}_{s,u}^{\mu,\nu}$. Similar as in the original MG, we also use $\widehat{V}_{s,u}^*$ to denote the NE value under the model $\widehat{\mathcal{G}}_{s,u}$, and use $\widehat{V}_{s,u}^{\mu,*}$ and $\widehat{V}_{s,u}^{*,\nu}$ to denote the best-response values of some given $\mu$ and $\nu$, under the model $\widehat{\mathcal{G}}_{s,u}$. Now we first have the following lemma based on Bernstein's inequality; see a similar argument in Lemma 5 in [23].

**Lemma A.5.** For fixed state $s$, action $(a, b)$, a finite set $U_s$, and $\delta > 0$, it holds that for all $u \in U_s$, with probability greater than $1 - \delta$,

$$\left|(P_{s,a,b} - \widehat{P}_{s,a,b}) \cdot \widehat{V}_{s,u}^*\right| \leq \sqrt{\frac{2\log(4|U_s|/\delta) \cdot \mathrm{Var}_{P_{s,a,b}}(\widehat{V}_{s,u}^*)}{N}} + \frac{2\log(4|U_s|/\delta)}{3(1-\gamma)N},$$

$$\left|(P_{s,a,b} - \widehat{P}_{s,a,b}) \cdot \widehat{V}_{s,u}^{\mu^*,*}\right| \leq \sqrt{\frac{2\log(4|U_s|/\delta) \cdot \mathrm{Var}_{P_{s,a,b}}(\widehat{V}_{s,u}^{\mu^*,*})}{N}} + \frac{2\log(4|U_s|/\delta)}{3(1-\gamma)N},$$

$$\left|(P_{s,a,b} - \widehat{P}_{s,a,b}) \cdot \widehat{V}_{s,u}^{*,\nu^*}\right| \leq \sqrt{\frac{2\log(4|U_s|/\delta) \cdot \mathrm{Var}_{P_{s,a,b}}(\widehat{V}_{s,u}^{*,\nu^*})}{N}} + \frac{2\log(4|U_s|/\delta)}{3(1-\gamma)N},$$

$$\left|(P_{s,a,b} - \widehat{P}_{s,a,b}) \cdot \widehat{V}_{s,u}^{\mu^*,\nu^*}\right| \leq \sqrt{\frac{2\log(4|U_s|/\delta) \cdot \mathrm{Var}_{P_{s,a,b}}(\widehat{V}_{s,u}^{\mu^*,\nu^*})}{N}} + \frac{2\log(4|U_s|/\delta)}{3(1-\gamma)N},$$

$$\left|(P_{s,a,b} - \widehat{P}_{s,a,b}) \cdot V^{\widehat{\mu}_{s,u},*}\right| \leq \sqrt{\frac{2\log(4|U_s|/\delta) \cdot \mathrm{Var}_{P_{s,a,b}}(V^{\widehat{\mu}_{s,u},*})}{N}} + \frac{2\log(4|U_s|/\delta)}{3(1-\gamma)N},$$

$$\left|(P_{s,a,b} - \widehat{P}_{s,a,b}) \cdot V^{*,\widehat{\nu}_{s,u}}\right| \leq \sqrt{\frac{2\log(4|U_s|/\delta) \cdot \mathrm{Var}_{P_{s,a,b}}(V^{*,\widehat{\nu}_{s,u}})}{N}} + \frac{2\log(4|U_s|/\delta)}{3(1-\gamma)N},$$

where $P_{s,a,b}$ and $\widehat{P}_{s,a,b}$ are the transition models extracted from the original game $\mathcal{G}$ and its empirical version $\widehat{\mathcal{G}}$, respectively (not related to either $\mathcal{G}_{s,u}$ or $\widehat{\mathcal{G}}_{s,u}$), and $(\widehat{\mu}_{s,a}, \widehat{\nu}_{s,a})$ is the output of the **Planning Oracle** using the auxiliary empirical model $\widehat{\mathcal{G}}_{s,u}$

*Proof.* The key observation is that the random variables $\widehat{P}_{s,a,b}$ and $\widehat{V}^*_{s,u}$ are independent. Using Bernstein's inequality along with a union bound over all $u \in U_s$, we obtain the first inequality. The other inequalities follow similarly, as $\widehat{P}_{s,a,b}$ is independent of $\widehat{V}^{\mu^*,*}_{s,u}$, $\widehat{V}^{*,\nu^*}_{s,u}$, $\widehat{V}^{\mu^*,\nu^*}_{s,u}$, $V^{\widehat{\mu}_{s,u},*}$, and $V^{*,\widehat{\nu}_{s,u}}$. This is because the latter terms are all decided by the original game $\mathcal{G}$, and/or the auxiliary empirical game $\widehat{\mathcal{G}}_{s,u}$ (not the original empirical game $\widehat{\mathcal{G}}$). $\qquad\square$

Note that the arguments in Lemma A.5 do not hold, if we replace $\widehat{V}^*_{s,u}$ by $\widehat{V}^*$, or $\widehat{V}^{\mu^*,*}_{s,u}$ by $\widehat{V}^{\mu^*,*}$, or $\widehat{V}^{*,\nu^*}_{s,u}$ by $\widehat{V}^{*,\nu^*}$. It will neither hold if we replace $\widehat{V}^{\mu^*,*}_{s,u}$ and $V^{\widehat{\mu}_{s,u},*}$ by some $\widehat{V}^{\mu,*}$ and $V^{\mu,*}$, for any $\mu$ that is dependent on $\widehat{P}$, e.g., the NE policy $\widehat{\mu}^*$ for the original empirical game $\widehat{\mathcal{G}}$. This is one of the key subtleties that is worth emphasizing.

Next we establish two lemmas that help guide the choices of $U_s$, so that $\widehat{V}^*_{s,u}$ (resp. $\widehat{V}^{\mu^*,*}_{s,u}$, $\widehat{V}^{*,\nu^*}_{s,u}$, and $\widehat{V}^{\mu^*,\nu^*}_{s,u}$) will be a good approximate of $\widehat{V}^*$ (resp. $\widehat{V}^{\mu^*,*}$, $\widehat{V}^{*,\nu^*}$, and $\widehat{V}^{\mu^*,\nu^*}$).

**Lemma A.6.** For the absorbing state $s$, and any joint policy $(\mu, \nu)$, suppose that $u^* = V^*_{\mathcal{G}}(s)$, $u^{\mu,*} = V^{\mu,*}_{\mathcal{G}}(s)$, $u^{*,\nu} = V^{*,\nu}_{\mathcal{G}}(s)$, and $u^{\mu,\nu} = V^{\mu,\nu}_{\mathcal{G}}(s)$. Then,

$$V^*_{\mathcal{G}} = V^*_{s,u^*} \qquad V^{\mu,*}_{\mathcal{G}} = V^{\mu,*}_{\mathcal{G}_{s,u^{\mu,*}}} \qquad V^{*,\nu}_{\mathcal{G}} = V^{*,\nu}_{\mathcal{G}_{s,u^{*,\nu}}} \qquad V^{\mu,\nu}_{\mathcal{G}} = V^{\mu,\nu}_{\mathcal{G}_{s,u^{\mu,\nu}}}.$$

*Proof.* For the first formula, we need to verify that $V^*_{\mathcal{G}}$ satisfies the optimal (Nash equilibrium) Bellman equation for the game $\mathcal{G}_{s,u^*}$. To this end, note that if $s' = s$, then $u^* = V^*_{\mathcal{G}}(s)$ satisfies the Bellman equation trivially, since $s$ is absorbing with the value $V^*_{s,u^*}(s) = u^*$.

On the other hand, for any $s' \neq s$, the outgoing transition model at $s'$ in $\mathcal{G}_{s,u^*}$ is the same as that in $\mathcal{G}$, and $V^*_{\mathcal{G}}(s')$ per se satisfies the Bellman equation in $\mathcal{G}$ (which are the same for $\mathcal{G}_{s,u^*}$ at these states $s' \neq s$). Thus, $V^*_{\mathcal{G}}$ satisfies the Bellman equation in $\mathcal{G}_{s,u^*}$ for all states. This proves the first equation. The proofs for the remaining three equations are analogous. $\qquad\square$

Perfect choices of $u$ have been specified in Lemma A.6 above. Moreover, we need to quantify how the value changes if we deviate from these perfect choices, i.e., the robustness to misspecification of $u$ [23]. This result is formally established in the following lemma; see also Lemma 7 in [23] for a similar result.

**Lemma A.7.** For any state $s$, $u, u' \in \mathbb{R}$, and joint policy pair $(\mu, \nu)$, we have

$$\left\|V^*_{s,u} - V^*_{s,u'}\right\|_\infty \leq |u - u'|, \qquad \left\|V^{\mu,*}_{s,u} - V^{\mu,*}_{s,u'}\right\|_\infty \leq |u - u'|,$$
$$\left\|V^{*,\nu}_{s,u} - V^{*,\nu}_{s,u'}\right\|_\infty \leq |u - u'|, \qquad \left\|V^{\mu,\nu}_{s,u} - V^{\mu,\nu}_{s,u'}\right\|_\infty \leq |u - u'|.$$

*Proof.* Note that $\|r_{s,u} - r_{s,u'}\|_\infty = (1-\gamma)|u - u'|$, since the reward functions only differ at $s$, where $r_{s,u}(s, a, b) = (1-\gamma)u$ and $r_{s,u'}(s, a, b) = (1-\gamma)u'$. We denote the NE policy pair in $\mathcal{G}_{s,u}$

by $(\mu_{s,u}^*, \nu_{s,u}^*)$. Thus,

$$Q_{s,u}^* - Q_{s,u'}^* = Q_{s,u}^{\mu_{s,u}^*, \nu_{s,u}^*} - Q_{s,u'}^{\mu_{s,u'}^*, \nu_{s,u'}^*} \leq Q_{s,u}^{\mu_{s,u}^*, \nu_{s,u'}^*} - Q_{s,u'}^{\mu_{s,u}^*, \nu_{s,u'}^*} \tag{A.17}$$

$$= \left(I - \gamma P_{s,u}^{\mu_{s,u}^*, \nu_{s,u'}^*}\right)^{-1} r_{s,u} - \left(I - \gamma P_{s,u'}^{\mu_{s,u}^*, \nu_{s,u'}^*}\right)^{-1} r_{s,u'} \tag{A.18}$$

$$= \left(I - \gamma P_{s,u}^{\mu_{s,u}^*, \nu_{s,u'}^*}\right)^{-1} \left(r_{s,u} - r_{s,u'}\right) \tag{A.19}$$

$$\leq \frac{\|r_{s,u} - r_{s,u'}\|_\infty}{1 - \gamma} = |u - u'|, \tag{A.20}$$

where (A.17) uses the fact that at the NE,

$$V_{s,u}^{\mu_{s,u}^*, \nu_{s,u}^*} = \min_\nu V_{s,u}^{\mu_{s,u}^*, \nu} \leq V_{s,u}^{\mu_{s,u}^*, \nu_{s,u'}^*}, \quad V_{s,u'}^{\mu_{s,u'}^*, \nu_{s,u'}^*} = \max_\mu V_{s,u'}^{\mu, \nu_{s,u'}^*} \geq V_{s,u'}^{\mu_{s,u}^*, \nu_{s,u'}^*},$$

implying the relationships of the corresponding Q-values; (A.18) is by definition; (A.19) uses the observation that $P_{s,u}^{\mu_{s,u}^*, \nu_{s,u'}^*}$ is the same as $P_{s,u'}^{\mu_{s,u}^*, \nu_{s,u'}^*}$ (transition is not affected by the value of $u$). Similarly, we can establish the lower bound that $Q_{s,u}^* - Q_{s,u'}^* \geq -|u - u'|$, which proves $\|Q_{s,u}^* - Q_{s,u'}^*\|_\infty \leq |u - u'|$. Moreover, we have

$$\left\|V_{s,u}^* - V_{s,u'}^*\right\|_\infty$$

$$= \left\| \max_{u \in \Delta(\mathcal{A})} \min_{\vartheta \in \Delta(\mathcal{B})} \mathbb{E}_{a \sim u, b \sim \vartheta}[Q_{s,u}^*(\cdot, a, b)] - \max_{u \in \Delta(\mathcal{A})} \min_{\vartheta \in \Delta(\mathcal{B})} \mathbb{E}_{a \sim u, b \sim \vartheta}[Q_{s,u'}^*(\cdot, a, b)] \right\|_\infty$$

$$\leq \max_{u \in \Delta(\mathcal{A}), \vartheta \in \Delta(\mathcal{B})} \left\| \mathbb{E}_{a \sim \mu(\cdot \mid s), b \sim \vartheta}[Q_{s,u}^*(\cdot, a, b)] - \mathbb{E}_{a \sim \mu(\cdot \mid s), b \sim \vartheta}[Q_{s,u'}^*(\cdot, a, b)] \right\|_\infty$$

$$\leq \left\|Q_{s,u}^* - Q_{s,u'}^*\right\|_\infty \leq |u - u'|,$$

which proves the first inequality.

For the second one, recalling that the best-response policy of $\mu$ under $\mathcal{G}_{s,u}$ being $\nu_{s,u}(\mu)$, we have

$$Q_{s,u}^{\mu,*} - Q_{s,u'}^{\mu,*} = \min_\nu Q_{s,u}^{\mu,\nu} - Q_{s,u'}^{\mu,*} = \min_\nu \left(I - \gamma P_{s,u}^{\mu,\nu}\right)^{-1} r_{s,u} - Q_{s,u'}^{\mu,*} \tag{A.21}$$

$$\leq \left(I - \gamma P_{s,u}^{\mu, \nu_{s,u'}(\mu)}\right)^{-1} r_{s,u} - \left(I - \gamma P_{s,u'}^{\mu, \nu_{s,u'}(\mu)}\right)^{-1} r_{s,u'} \tag{A.22}$$

$$= \left(I - \gamma P_{s,u}^{\mu, \nu_{s,u'}(\mu)}\right)^{-1} \left(r_{s,u} - r_{s,u'}\right) \leq \frac{\|r_{s,u} - r_{s,u'}\|_\infty}{1 - \gamma} = |u - u'|, \tag{A.23}$$

where (A.21) uses the definition of a best-response value, (A.22) plugs in the best-response policy $\nu_{s,u'}(\mu)$, and (A.23) also uses the fact that the transition does not depend on the value $u$. A lower bound can be established by noticing that $Q_{s,u'}^{\mu,*} = \min_\nu Q_{s,u'}^{\mu,\nu} \leq Q_{s,u'}^{\mu, \nu_{s,u}(\mu)}$. This proves $\|Q_{s,u}^{\mu,*} - Q_{s,u'}^{\mu,*}\|_\infty \leq |u - u'|$. Furthermore, notice that

$$\left\|V_{s,u}^{\mu,*} - V_{s,u'}^{\mu,*}\right\|_\infty = \left\| \min_{\vartheta \in \Delta(\mathcal{B})} \mathbb{E}_{a \sim \mu(\cdot \mid s), b \sim \vartheta}[Q_{s,u}^{\mu,*}(\cdot, a, b)] - \min_{\vartheta \in \Delta(\mathcal{B})} \mathbb{E}_{a \sim \mu(\cdot \mid s), b \sim \vartheta}[Q_{s,u'}^{\mu,*}(\cdot, a, b)] \right\|_\infty$$

$$\leq \max_{\vartheta \in \Delta(\mathcal{B})} \left\| \mathbb{E}_{a \sim \mu(\cdot \mid s), b \sim \vartheta}[Q_{s,u}^{\mu,*}(\cdot, a, b)] - \mathbb{E}_{a \sim \mu(\cdot \mid s), b \sim \vartheta}[Q_{s,u'}^{\mu,*}(\cdot, a, b)] \right\|_\infty$$

$$\leq \left\|Q_{s,u}^{\mu,*} - Q_{s,u'}^{\mu,*}\right\|_\infty \leq |u - u'|,$$

which proves the second inequality. Similar arguments can also be used to establish the third and the fourth inequalities. This completes the proof. $\qquad\square$

We are now ready to show the main result in this section.

**Lemma A.8.** For any state $s$, joint action pair $(a, b)$, and a finite set $U_s$, define

$$\Gamma_{U_s, \delta, N} := \frac{2 \log(4|U_s|/\delta)}{3(1 - \gamma)N}, \qquad \Upsilon_{U_s, \delta, N} := 2 + \sqrt{\frac{2 \log(4|U_s|/\delta)}{N}}.$$

Then, with probability greater than $1 - \delta$, we have

$$\left|(P_{s,a,b} - \widehat{P}_{s,a,b})\widehat{V}^*\right| \leq \sqrt{\frac{2\log(4|U_s|/\delta)\operatorname{Var}_{P_{s,a,b}}(\widehat{V}^*)}{N}} + \Gamma_{U_s,\delta,N} + \min_{u \in U_s}\left|\widehat{V}^*(s) - u\right| \cdot \Upsilon_{U_s,\delta,N}$$

$$\left|(P_{s,a,b} - \widehat{P}_{s,a,b})\widehat{V}^{\mu^*,*}\right| \leq \sqrt{\frac{2\log(4|U_s|/\delta)\operatorname{Var}_{P_{s,a,b}}(\widehat{V}^{\mu^*,*})}{N}} + \Gamma_{U_s,\delta,N}$$
$$+ \min_{u \in U_s}\left|\widehat{V}^{\mu^*,*}(s) - u\right| \cdot \Upsilon_{U_s,\delta,N}$$

$$\left|(P_{s,a,b} - \widehat{P}_{s,a,b})\widehat{V}^{*,\nu^*}\right| \leq \sqrt{\frac{2\log(4|U_s|/\delta)\operatorname{Var}_{P_{s,a,b}}(\widehat{V}^{*,\nu^*})}{N}} + \Gamma_{U_s,\delta,N}$$
$$+ \min_{u \in U_s}\left|\widehat{V}^{*,\nu^*}(s) - u\right| \cdot \Upsilon_{U_s,\delta,N}$$

$$\left|(P_{s,a,b} - \widehat{P}_{s,a,b})\widehat{V}^{\mu^*,\nu^*}\right| \leq \sqrt{\frac{2\log(4|U_s|/\delta)\operatorname{Var}_{P_{s,a,b}}(\widehat{V}^{\mu^*,\nu^*})}{N}} + \Gamma_{U_s,\delta,N}$$
$$+ \min_{u \in U_s}\left|\widehat{V}^{\mu^*,\nu^*}(s) - u\right| \cdot \Upsilon_{U_s,\delta,N}.$$

Moreover, recalling that $(\widehat{\mu}_{s,u}, \widehat{\nu}_{s,u})$ is the output of the **Planning Oracle** using $\widehat{\mathcal{G}}_{s,u}$, we have

$$\left|(P_{s,a,b} - \widehat{P}_{s,a,b})V^{\widehat{\mu},*}\right| \leq \sqrt{\frac{2\log(4|U_s|/\delta) \cdot \operatorname{Var}_{P_{s,a,b}}(V^{\widehat{\mu},*})}{N}} + \Gamma_{U_s,\delta,N}$$
$$+ \min_{u \in U_s}\left\|V^{\widehat{\mu},*} - V^{\widehat{\mu}_{s,u},*}\right\|_\infty \cdot \Upsilon_{U_s,\delta,N},$$

$$\left|(P_{s,a,b} - \widehat{P}_{s,a,b})V^{*,\widehat{\nu}}\right| \leq \sqrt{\frac{2\log(4|U_s|/\delta) \cdot \operatorname{Var}_{P_{s,a,b}}(V^{*,\widehat{\nu}})}{N}} + \Gamma_{U_s,\delta,N}$$
$$+ \min_{u \in U_s}\left\|V^{*,\widehat{\nu}} - V^{*,\widehat{\nu}_{s,u}}\right\|_\infty \cdot \Upsilon_{U_s,\delta,N}.$$

*Proof.* First, for all $u \in U_s$ and with probability greater than $1 - \delta$, we have

$$\left|(P_{s,a,b} - \widehat{P}_{s,a,b})\widehat{V}^*\right| = \left|(P_{s,a,b} - \widehat{P}_{s,a,b})(\widehat{V}^* - \widehat{V}^*_{s,u} + \widehat{V}^*_{s,u})\right|$$

$$\leq \left|(P_{s,a,b} - \widehat{P}_{s,a,b})(\widehat{V}^* - \widehat{V}^*_{s,u})\right| + \left|(P_{s,a,b} - \widehat{P}_{s,a,b})\widehat{V}^*_{s,u}\right| \tag{A.24}$$

$$\leq 2 \cdot \left\|\widehat{V}^* - \widehat{V}^*_{s,u}\right\|_\infty + \left|(P_{s,a,b} - \widehat{P}_{s,a,b})\widehat{V}^*_{s,u}\right| \tag{A.25}$$

$$\leq 2 \cdot \left\|\widehat{V}^* - \widehat{V}^*_{s,u}\right\|_\infty + \sqrt{\frac{2\log(4|U_s|/\delta) \cdot \operatorname{Var}_{P_{s,a,b}}(\widehat{V}^*_{s,u})}{N}} + \frac{2\log(4|U_s|/\delta)}{3(1-\gamma)N} \tag{A.26}$$

$$\leq \left\|\widehat{V}^* - \widehat{V}^*_{s,u}\right\|_\infty \left(2 + \sqrt{\frac{2\log(4|U_s|/\delta)}{N}}\right)$$

$$+ \sqrt{\frac{2\log(4|U_s|/\delta) \cdot \operatorname{Var}_{P_{s,a,b}}(\widehat{V}^*)}{N}} + \frac{2\log(4|U_s|/\delta)}{3(1-\gamma)N} \tag{A.27}$$

where (A.24)-(A.25) use triangle inequality, (A.26) is due to Lemma A.5, and (A.27) uses the facts that $\sqrt{\operatorname{Var}_{P_{s,a,b}}(X + Y)} \leq \sqrt{\operatorname{Var}_{P_{s,a,b}}(X)} + \sqrt{\operatorname{Var}_{P_{s,a,b}}(Y)}$, and $\sqrt{\operatorname{Var}_{P_{s,a,b}}(X)} \leq \|X\|_\infty$. Moreover, by Lemmas A.6 and A.7, we obtain that

$$\left\|\widehat{V}^* - \widehat{V}^*_{s,u}\right\|_\infty = \left\|\widehat{V}^*_{s,\widehat{V}^*(s)} - \widehat{V}^*_{s,u}\right\|_\infty \leq \left|\widehat{V}^*(s) - u\right|,$$

which, combined with (A.27) and taken minimization over all $u \in U_s$, yields the first inequality. Proofs for the remaining inequalities are analogous, except that for the last two, the norms $\|V^{\widehat{\mu},*} - V^{\widehat{\mu}_{s,u},*}\|_\infty$ and $\|V^{*,\widehat{\nu}} - V^{*,\widehat{\nu}_{s,u}}\|_\infty$ are kept and not further bounded. $\square$

Next we establish the important result that characterizes the errors $|(P - \widehat{P})\widehat{V}^*|$, $|(P - \widehat{P})\widehat{V}^{\mu^*,*}|$, $|(P - \widehat{P})\widehat{V}^{*,\nu^*}|$, and $|(P - \widehat{P})\widehat{V}^{\mu^*,\nu^*}|$, which could not have been handled without the arguments above, due to the dependence between $\widehat{P}$ and $\widehat{V}^*$ (and also $\widehat{V}^{\mu^*,*}$, $\widehat{V}^{*,\nu^*}$, and $\widehat{V}^{\mu^*,\nu^*}$).

**Lemma A.9.** For any $\delta \in (0,1]$, with probability greater than $1 - \delta$, it holds that

$$\left|(P - \widehat{P})\widehat{V}^*\right| \leq \sqrt{\frac{2\log\left(16|\mathcal{S}||\mathcal{A}||\mathcal{B}|/[(1-\gamma)^2\delta]\right) \cdot \mathrm{Var}_P(\widehat{V}^*)}{N}} + \Delta'_{\delta,N}$$

$$\left|(P - \widehat{P})\widehat{V}^{\mu^*,*}\right| \leq \sqrt{\frac{2\log\left(16|\mathcal{S}||\mathcal{A}||\mathcal{B}|/[(1-\gamma)^2\delta]\right) \cdot \mathrm{Var}_P(\widehat{V}^{\mu^*,*})}{N}} + \Delta'_{\delta,N}$$

$$\left|(P - \widehat{P})\widehat{V}^{*,\nu^*}\right| \leq \sqrt{\frac{2\log\left(16|\mathcal{S}||\mathcal{A}||\mathcal{B}|/[(1-\gamma)^2\delta]\right) \cdot \mathrm{Var}_P(\widehat{V}^{*,\nu^*})}{N}} + \Delta'_{\delta,N}$$

$$\left|(P - \widehat{P})\widehat{V}^{\mu^*,\nu^*}\right| \leq \sqrt{\frac{2\log\left(16|\mathcal{S}||\mathcal{A}||\mathcal{B}|/[(1-\gamma)^2\delta]\right) \cdot \mathrm{Var}_P(\widehat{V}^{\mu^*,\nu^*})}{N}} + \Delta'_{\delta,N}$$

where $\Delta'_{\delta,N}$ is defined as

$$\Delta'_{\delta,N} = \sqrt{\frac{c\log\left(c|\mathcal{S}||\mathcal{A}||\mathcal{B}|/[(1-\gamma)^2\delta]\right)}{N}} + \frac{c\log\left(c|\mathcal{S}||\mathcal{A}||\mathcal{B}|/[(1-\gamma)^2\delta]\right)}{(1-\gamma)N},$$

and $c$ is some absolute constant.

*Proof.* Let $U_s$ denote a set with evenly spaced elements in the interval $[V^*(s) - \Delta_{\delta/2,N}, V^*(s) + \Delta_{\delta/2,N}]$, with $|U_s| = 2/(1-\gamma)^2$, and $\Delta_{\delta,N}$ being defined in Lemma A.4. Lemma A.4 shows that with probability greater than $1 - \delta/2$,

$$\widehat{V}^*(s) \in \left[V^*(s) - \Delta_{\delta/2,N}, \ V^*(s) + \Delta_{\delta/2,N}\right] \tag{A.28}$$

for all $s \in \mathcal{S}$. Since each subinterval determined by $U_s$ is of length $2\Delta_{\delta/2,N}/(|U_s| - 1)$, and $\widehat{V}^*(s)$ will fall into one of them, we know that

$$\min_{u \in U_s} \left|\widehat{V}^*(s) - u\right| \leq \frac{2\Delta_{\delta/2,N}}{|U_s| - 1} = \frac{2\gamma}{(|U_s| - 1)(1-\gamma)^2}\sqrt{\frac{2\log(4|\mathcal{S}||\mathcal{A}||\mathcal{B}|/\delta)}{N}}$$

$$\leq 2\gamma\sqrt{\frac{2\log(4|\mathcal{S}||\mathcal{A}||\mathcal{B}|/\delta)}{N}},$$

where we have used the fact that $|U_s| \geq 1/(1-\gamma)^2 + 1$. We then choose $\delta/2$ to be $\delta/(2|\mathcal{S}||\mathcal{A}||\mathcal{B}|)$ in Lemma A.8, so that it holds for all states and joint actions with probability greater than $1 - \delta/2$. By substitution and noting that the two events in Lemmas A.4 and A.8 both fail with probability $\delta/2$, we obtain the first inequality by properly choosing the constant $c$. Similarly, for the other two inequalities, note that Lemma A.4 can be applied to show that $\widehat{V}^{\mu^*,*}(s)$, $\widehat{V}^{*,\nu^*}(s)$, and $\widehat{V}^{\mu^*,\nu^*}(s)$, all lie in the interval in (A.28) (centered at $V^*(s)$). By similar arguments, the remaining three inequalities can be proved (note that Lemma A.8 can be applied to $\widehat{V}^{\mu^*,*}(s)$, $\widehat{V}^{*,\nu^*}(s)$, and $\widehat{V}^{\mu^*,\nu^*}(s)$, as well). $\qquad\square$

Lastly, with a smooth **Planning Oracle**, see Definition 3.4, we can similarly establish the following error bounds on $|(P - \widehat{P})V^{\widehat{\mu},*}|$ and $|(P - \widehat{P})V^{*,\widehat{\nu}}|$, thanks to Lemma A.8.

**Lemma A.10.** With a smooth **Planning Oracle** that has smooth constant $C$ (see Definition 3.4), for any $\delta \in (0,1]$, with probability greater than $1 - \delta$, it holds that

$$\left|(P - \widehat{P})V^{\widehat{\mu},*}\right| \leq \sqrt{\frac{2\log\left(8(C+1)|\mathcal{S}||\mathcal{A}||\mathcal{B}|/[(1-\gamma)^4\delta]\right) \cdot \mathrm{Var}_P(V^{\widehat{\mu},*})}{N}} + \Delta''_{\delta,N}$$

$$\left|(P - \widehat{P})V^{*,\widehat{\nu}}\right| \leq \sqrt{\frac{2\log\left(8(C+1)|\mathcal{S}||\mathcal{A}||\mathcal{B}|/[(1-\gamma)^4\delta]\right) \cdot \mathrm{Var}_P(V^{*,\widehat{\nu}})}{N}} + \Delta''_{\delta,N}$$

where $\Delta''_{\delta,N}$ is defined as

$$\Delta''_{\delta,N} = \sqrt{\frac{c\log\left(c(C+1)|\mathcal{S}||\mathcal{A}||\mathcal{B}|/[(1-\gamma)^4\delta]\right)}{N}} + \frac{c\log\left(c(C+1)|\mathcal{S}||\mathcal{A}||\mathcal{B}|/[(1-\gamma)^4\delta]\right)}{(1-\gamma)N},$$

for some absolute constant $c$.

*Proof.* Following the proof of Lemma A.9, let $U_s$ denote a set with evenly spaced elements in the interval $[V^*(s) - \Delta_{\delta/2,N}, V^*(s) + \Delta_{\delta/2,N}]$, with $\Delta_{\delta,N}$ being defined in Lemma A.4. By Lemma A.4, we know that $\widehat{V}^*(s)$ lies in this interval with probability greater than $1 - \delta/2$, for all $s \in \mathcal{S}$. Now we choose $|U_s| = (C+1)/(1-\gamma)^4$, where $C$ is the smooth coefficient in Definition 3.4. As $\widehat{V}^*(s)$ will fall into one of the subintervals determined by $U_s$, we have

$$\min_{u \in U_s} \left| \widehat{V}^*(s) - u \right| \leq \frac{2\Delta_{\delta/2,N}}{|U_s| - 1} \leq \frac{2\gamma(1-\gamma)^2}{C} \cdot \sqrt{\frac{2\log(4|\mathcal{S}||\mathcal{A}||\mathcal{B}|/\delta)}{N}}, \tag{A.29}$$

which also uses the fact $|U_s| \geq C/(1-\gamma)^4 + 1$. Furthermore, by Definition 3.4 and the proof of Lemma A.7, we have

$$\left\| \widehat{\mu} - \widehat{\mu}_{s,u} \right\|_{TV} \leq C \cdot \|\widehat{Q}^* - \widehat{Q}^*_{s,u}\|_\infty \leq C \cdot \left| \widehat{V}^*(s) - u \right|. \tag{A.30}$$

On the other hand, we have

$$\left\| V^{\widehat{\mu},*} - V^{\widehat{\mu}_{s,u},*} \right\|_\infty \leq \max_{\vartheta \in \Delta(\mathcal{B})} \left\| \mathbb{E}_{a \sim \widehat{\mu}(\cdot \mid s), b \sim \vartheta}[Q^{\widehat{\mu},*}(\cdot, a, b)] - \mathbb{E}_{a \sim \widehat{\mu}_{s,u}(\cdot \mid s), b \sim \vartheta}[Q^{\widehat{\mu}_{s,u},*}(\cdot, a, b)] \right\|_\infty$$

$$\leq \max_{\vartheta \in \Delta(\mathcal{B})} \left\| \mathbb{E}_{a \sim \widehat{\mu}(\cdot \mid s), b \sim \vartheta}[Q^{\widehat{\mu},*}(\cdot, a, b)] - \mathbb{E}_{a \sim \widehat{\mu}(\cdot \mid s), b \sim \vartheta}[Q^{\widehat{\mu}_{s,u},*}(\cdot, a, b)] \right\|_\infty$$

$$+ \max_{\vartheta \in \Delta(\mathcal{B})} \left\| \mathbb{E}_{a \sim \widehat{\mu}(\cdot \mid s), b \sim \vartheta}[Q^{\widehat{\mu}_{s,u},*}(\cdot, a, b)] - \mathbb{E}_{a \sim \widehat{\mu}_{s,u}(\cdot \mid s), b \sim \vartheta}[Q^{\widehat{\mu}_{s,u},*}(\cdot, a, b)] \right\|_\infty$$

$$\leq \left\| Q^{\widehat{\mu},*} - Q^{\widehat{\mu}_{s,u},*} \right\|_\infty + \left\| \widehat{\mu} - \widehat{\mu}_{s,u} \right\|_{TV} \cdot \left\| Q^{\widehat{\mu}_{s,u},*} \right\|_\infty \tag{A.31}$$

$$\leq \gamma \left\| V^{\widehat{\mu},*} - V^{\widehat{\mu}_{s,u},*} \right\|_\infty + \frac{C}{1-\gamma} \cdot \left| \widehat{V}^*(s) - u \right|, \tag{A.32}$$

where (A.31) uses Hölder's inequality, and (A.32) follows by expanding the Q-value functions, using (A.30), and noticing that $\|Q^{\widehat{\mu}_{s,u},*}\|_\infty \leq 1/(1-\gamma)$. Combining (A.32) and (A.29), and taking $\min$ over $u \in U_s$, we have

$$\min_{u \in U_s} \left\| V^{\widehat{\mu},*} - V^{\widehat{\mu}_{s,u},*} \right\|_\infty \leq \frac{C}{(1-\gamma)^2} \cdot \min_{u \in U_s} \left| \widehat{V}^*(s) - u \right| \leq 2\gamma \cdot \sqrt{\frac{2\log(4|\mathcal{S}||\mathcal{A}||\mathcal{B}|/\delta)}{N}}.$$

The rest of the proof follows the arguments of Lemma A.9, which combines the last two inequalities in Lemma A.8 to obtain the desired bound. Note that the absolute constant here might be different from that in Lemma A.9. The proof for the second inequality is analogous. □

## A.3 Proof of Theorem 3.2

We are now ready to prove Theorem 3.2. To this end, we first establish the following lemma.

**Lemma A.11.** For any policy pair $(\widehat{\mu}, \widehat{\nu})$ that satisfies the condition in Theorem 3.2, there exists some absolute constant $c$ such that

$$\left\| Q^{\widehat{\mu},\widehat{\nu}} - \widehat{Q}^{\widehat{\mu},\widehat{\nu}} \right\|_\infty \leq \frac{\gamma}{1-\alpha_{\delta,N}} \left( \sqrt{\frac{c\log(c|\mathcal{S}||\mathcal{A}||\mathcal{B}|/[(1-\gamma)^2\delta])}{(1-\gamma)^3 N}} + \frac{c\log(c|\mathcal{S}||\mathcal{A}||\mathcal{B}|/[(1-\gamma)^2\delta])}{(1-\gamma)^2 N} \right)$$

$$+ \frac{1}{1-\alpha_{\delta,N}} \cdot \frac{\gamma\epsilon_{opt}}{(1-\gamma)} \left( 1 + \sqrt{\frac{\log(c|\mathcal{S}||\mathcal{A}||\mathcal{B}|/[(1-\gamma)^2\delta])}{N}} \right)$$

$$\left\| Q^* - \widehat{Q}^{\mu^*,*} \right\|_\infty \leq \frac{\gamma}{1-\alpha_{\delta,N}} \left( \sqrt{\frac{c\log(c|\mathcal{S}||\mathcal{A}||\mathcal{B}|/[(1-\gamma)^2\delta])}{(1-\gamma)^3 N}} + \frac{c\log(c|\mathcal{S}||\mathcal{A}||\mathcal{B}|/[(1-\gamma)^2\delta])}{(1-\gamma)^2 N} \right)$$

$$\left\| Q^* - \widehat{Q}^{*,\nu^*} \right\|_\infty \leq \frac{\gamma}{1-\alpha_{\delta,N}} \left( \sqrt{\frac{c\log(c|\mathcal{S}||\mathcal{A}||\mathcal{B}|/[(1-\gamma)^2\delta])}{(1-\gamma)^3 N}} + \frac{c\log(c|\mathcal{S}||\mathcal{A}||\mathcal{B}|/[(1-\gamma)^2\delta])}{(1-\gamma)^2 N} \right),$$

where

$$\alpha_{\delta,N} = \frac{\gamma}{1-\gamma} \sqrt{\frac{2\log(16|\mathcal{S}||\mathcal{A}||\mathcal{B}|/[(1-\gamma)^2\delta])}{N}}.$$

*Proof.* Note that

$$\|Q^{\widehat{\mu},\widehat{\nu}} - \widehat{Q}^{\widehat{\mu},\widehat{\nu}}\|_\infty = \gamma\big\|(I - \gamma P^{\widehat{\mu},\widehat{\nu}})^{-1}(P - \widehat{P})\widehat{V}^{\widehat{\mu},\widehat{\nu}}\big\|_\infty \tag{A.33}$$

$$\leq \gamma\big\|(I - \gamma P^{\widehat{\mu},\widehat{\nu}})^{-1}(P - \widehat{P})\widehat{V}^*\big\|_\infty + \gamma\big\|(I - \gamma P^{\widehat{\mu},\widehat{\nu}})^{-1}(P - \widehat{P})(\widehat{V}^{\widehat{\mu},\widehat{\nu}} - \widehat{V}^*)\big\|_\infty \tag{A.34}$$

$$\leq \gamma\big\|(I - \gamma P^{\widehat{\mu},\widehat{\nu}})^{-1}\big|(P - \widehat{P})\widehat{V}^*\big|\big\|_\infty + \frac{2\gamma\epsilon_{opt}}{1 - \gamma}, \tag{A.35}$$

where (A.33) is due to Lemma A.1; (A.34) uses triangle inequality; and (A.35) is due to the non-negativeness of the entries in $(I - \gamma P^{\widehat{\mu},\widehat{\nu}})^{-1}$, the sub-optimality of $(\widehat{\mu}, \widehat{\nu})$, and Lemma A.2. Since the first term in (A.35) can be bounded using Lemma A.9, we have

$$\|Q^{\widehat{\mu},\widehat{\nu}} - \widehat{Q}^{\widehat{\mu},\widehat{\nu}}\|_\infty$$

$$\leq \gamma\sqrt{\frac{2\log\left(16|\mathcal{S}||\mathcal{A}||\mathcal{B}|/[(1-\gamma)^2\delta]\right)}{N}}\left\|(I - \gamma P^{\widehat{\mu},\widehat{\nu}})^{-1}\sqrt{\mathrm{Var}_P(\widehat{V}^*)}\right\|_\infty + \frac{\gamma\Delta'_{\delta,N}}{1-\gamma} + \frac{2\gamma\epsilon_{opt}}{1-\gamma}$$

$$\leq \gamma\sqrt{\frac{2\log\left(16|\mathcal{S}||\mathcal{A}||\mathcal{B}|/[(1-\gamma)^2\delta]\right)}{N}}\left\|(I - \gamma P^{\widehat{\mu},\widehat{\nu}})^{-1}\left(\sqrt{\mathrm{Var}_P(V^{\widehat{\mu},\widehat{\nu}})} + \sqrt{\mathrm{Var}_P(V^{\widehat{\mu},\widehat{\nu}} - \widehat{V}^{\widehat{\mu},\widehat{\nu}})}\right)\right\|_\infty$$

$$+ \gamma\sqrt{\frac{2\log\left(16|\mathcal{S}||\mathcal{A}||\mathcal{B}|/[(1-\gamma)^2\delta]\right)}{N}}\left\|(I - \gamma P^{\widehat{\mu},\widehat{\nu}})^{-1}\left(\sqrt{\mathrm{Var}_P(\widehat{V}^{\widehat{\mu},\widehat{\nu}} - \widehat{V}^*)}\right)\right\|_\infty$$

$$+ \frac{\gamma\Delta'_{\delta,N}}{1-\gamma} + \frac{2\gamma\epsilon_{opt}}{1-\gamma} \tag{A.36}$$

$$\leq \gamma\sqrt{\frac{2\log\left(16|\mathcal{S}||\mathcal{A}||\mathcal{B}|/[(1-\gamma)^2\delta]\right)}{N}}\left(\sqrt{\frac{2}{(1-\gamma)^3}} + \frac{\|V^{\widehat{\mu},\widehat{\nu}} - \widehat{V}^{\widehat{\mu},\widehat{\nu}}\|_\infty}{1-\gamma} + \frac{\epsilon_{opt}}{1-\gamma}\right)$$

$$+ \frac{\gamma\Delta'_{\delta,N}}{1-\gamma} + \frac{2\gamma\epsilon_{opt}}{1-\gamma} \tag{A.37}$$

$$\leq \gamma\sqrt{\frac{2\log\left(16|\mathcal{S}||\mathcal{A}||\mathcal{B}|/[(1-\gamma)^2\delta]\right)}{N}}\left(\sqrt{\frac{2}{(1-\gamma)^3}} + \frac{\|Q^{\widehat{\mu},\widehat{\nu}} - \widehat{Q}^{\widehat{\mu},\widehat{\nu}}\|_\infty}{1-\gamma}\right) + \frac{\gamma\Delta'_{\delta,N}}{1-\gamma}$$

$$+ \left(2 + \sqrt{\frac{2\log\left(16|\mathcal{S}||\mathcal{A}||\mathcal{B}|/[(1-\gamma)^2\delta]\right)}{N}}\right) \cdot \frac{\gamma\epsilon_{opt}}{1-\gamma}, \tag{A.38}$$

where (A.36) uses the fact that $\sqrt{\mathrm{Var}_P(X+Y)} \leq \sqrt{\mathrm{Var}_P(X)} + \sqrt{\mathrm{Var}_P(Y)}$; (A.37) is due to Lemma A.3, the fact that $\sqrt{\mathrm{Var}_P(V^{\widehat{\mu},\widehat{\nu}} - \widehat{V}^{\widehat{\mu},\widehat{\nu}})} \leq \|V^{\widehat{\mu},\widehat{\nu}} - \widehat{V}^{\widehat{\mu},\widehat{\nu}}\|_\infty$, and $\|\widehat{V}^{\widehat{\mu},\widehat{\nu}} - \widehat{V}^*\|_\infty \leq \epsilon_{opt}$; (A.38) is due to $\|V^{\widehat{\mu},\widehat{\nu}} - \widehat{V}^{\widehat{\mu},\widehat{\nu}}\|_\infty \leq \|Q^{\widehat{\mu},\widehat{\nu}} - \widehat{Q}^{\widehat{\mu},\widehat{\nu}}\|_\infty$. Solving for $\|Q^{\widehat{\mu},\widehat{\nu}} - \widehat{Q}^{\widehat{\mu},\widehat{\nu}}\|_\infty$ in (A.38) yields the desired inequality.

For the second inequality, by Lemma A.1, we first have

$$\underbrace{\gamma(I - \gamma P^{\mu^*,\nu^*})^{-1}(P - \widehat{P})\widehat{V}^{\mu^*,\nu^*}}_{Q^{\mu^*,\nu^*} - \widehat{Q}^{\mu^*,\nu^*}} \leq Q^* - \widehat{Q}^{\mu^*,*} \leq \underbrace{\gamma(I - \gamma P^{\mu^*,\widehat{\nu(\mu^*)}})^{-1}(P - \widehat{P})\widehat{V}^{\mu^*,*}}_{Q^{\mu^*,\widehat{\nu(\mu^*)}} - \widehat{Q}^{\mu^*,*}}.$$

Thus, we obtain that

$$\|Q^* - \widehat{Q}^{\mu^*,*}\|_\infty \leq \max\left\{\|Q^{\mu^*,\nu^*} - \widehat{Q}^{\mu^*,\nu^*}\|_\infty, \; \|Q^{\mu^*,\widehat{\nu(\mu^*)}} - \widehat{Q}^{\mu^*,*}\|_\infty\right\}$$

$$= \max\left\{\gamma\|(I - \gamma P^{\mu^*,\nu^*})^{-1}(P - \widehat{P})\widehat{V}^{\mu^*,\nu^*}\|_\infty, \; \gamma\|(I - \gamma P^{\mu^*,\widehat{\nu(\mu^*)}})^{-1}(P - \widehat{P})\widehat{V}^{\mu^*,*}\|_\infty\right\}. \tag{A.39}$$

For the first term in the $\max$ operator above, by similar arguments from (A.36)-(A.38), we have

$$\left\|Q^{\mu^*,\nu^*} - \widehat{Q}^{\mu^*,\nu^*}\right\|_\infty = \gamma\left\|(I - \gamma P^{\mu^*,\nu^*})^{-1}(P - \widehat{P})\widehat{V}^{\mu^*,\nu^*}\right\|_\infty$$

$$\leq \gamma\sqrt{\frac{2\log\left(16|\mathcal{S}||\mathcal{A}||\mathcal{B}|/[(1-\gamma)^2\delta]\right)}{N}}\left\|(I - \gamma P^{\mu^*,\nu^*})^{-1}\sqrt{\mathrm{Var}_P(\widehat{V}^{\mu^*,\nu^*})}\right\|_\infty + \frac{\gamma\Delta'_{\delta,N}}{1-\gamma}$$
(A.40)

$$\leq \gamma\sqrt{\frac{2\log\left(16|\mathcal{S}||\mathcal{A}||\mathcal{B}|/[(1-\gamma)^2\delta]\right)}{N}}\left\|(I - \gamma P^{\mu^*,\nu^*})^{-1}\sqrt{\mathrm{Var}_P(V^{\mu^*,\nu^*} - \widehat{V}^{\mu^*,\nu^*})}\right\|_\infty$$

$$+ \gamma\sqrt{\frac{2\log\left(16|\mathcal{S}||\mathcal{A}||\mathcal{B}|/[(1-\gamma)^2\delta]\right)}{N}}\left\|(I - \gamma P^{\mu^*,\nu^*})^{-1}\sqrt{\mathrm{Var}_P(V^{\mu^*,\nu^*})}\right\|_\infty + \frac{\gamma\Delta'_{\delta,N}}{1-\gamma}$$
(A.41)

$$\leq \gamma\sqrt{\frac{2\log\left(16|\mathcal{S}||\mathcal{A}||\mathcal{B}|/[(1-\gamma)^2\delta]\right)}{N}} \cdot \frac{\left\|Q^{\mu^*,\nu^*} - \widehat{Q}^{\mu^*,\nu^*}\right\|_\infty}{1-\gamma}$$

$$+ \gamma\sqrt{\frac{2\log\left(16|\mathcal{S}||\mathcal{A}||\mathcal{B}|/[(1-\gamma)^2\delta]\right)}{N}} \cdot \sqrt{\frac{2}{(1-\gamma)^3}} + \frac{\gamma\Delta'_{\delta,N}}{1-\gamma},$$
(A.42)

where (A.40) is due to Lemma A.9, (A.41) uses triangle inequality, and (A.43) uses Lemma A.3. Solving for $\left\|Q^{\mu^*,\nu^*} - \widehat{Q}^{\mu^*,\nu^*}\right\|_\infty$ gives the bound for it.

Similarly, the second term in the $\max$ operator in (A.39) can be bounded by

$$\left\|Q^{\mu^*,\widehat{\nu(\mu^*)}} - \widehat{Q}^{\mu^*,*}\right\|_\infty \leq \gamma\sqrt{\frac{2\log\left(16|\mathcal{S}||\mathcal{A}||\mathcal{B}|/[(1-\gamma)^2\delta]\right)}{N}} \cdot \frac{\left\|Q^{\mu^*,\widehat{\nu(\mu^*)}} - \widehat{Q}^{\mu^*,*}\right\|_\infty}{1-\gamma}$$

$$+ \gamma\sqrt{\frac{2\log\left(16|\mathcal{S}||\mathcal{A}||\mathcal{B}|/[(1-\gamma)^2\delta]\right)}{N}} \cdot \sqrt{\frac{2}{(1-\gamma)^3}} + \frac{\gamma\Delta'_{\delta,N}}{1-\gamma},$$
(A.43)

which can be solved to obtain a bound for $\left\|Q^{\mu^*,\widehat{\nu(\mu^*)}} - \widehat{Q}^{\mu^*,*}\right\|_\infty$. Combining the two bounds and (A.39), we prove the second inequality in the lemma. The proof for the third inequality is analogous. $\square$

With Lemma A.11 in hand, we are now ready to prove Theorem 3.2. Note that the condition on $N$ in Theorem 3.2 makes $\alpha_{\delta,N} < 1/2$. Thus, by (A.2)-(A.3) in Lemma A.1 with $(\mu,\nu)$ being replaced by $(\widehat{\mu},\widehat{\nu})$, we have

$$-\|Q^{\widehat{\mu},\widehat{\nu}} - \widehat{Q}^{\widehat{\mu},\widehat{\nu}}\|_\infty - \gamma\epsilon_{opt} - \|\widehat{Q}^{\mu^*,*} - Q^*\|_\infty \leq Q^{\widehat{\mu},\widehat{\nu}} - Q^*$$

$$\leq \|Q^{\widehat{\mu},\widehat{\nu}} - \widehat{Q}^{\widehat{\mu},\widehat{\nu}}\|_\infty + \gamma\epsilon_{opt} + \|\widehat{Q}^{*,\nu^*} - Q^*\|_\infty,$$

where we use

$$\|\widehat{Q}^{\widehat{\mu},\widehat{\nu}} - \widehat{Q}^*\|_\infty = \gamma\|P\widehat{V}^{\widehat{\mu},\widehat{\nu}} - P\widehat{V}^*\|_\infty \leq \gamma\|\widehat{V}^{\widehat{\mu},\widehat{\nu}} - \widehat{V}^*\|_\infty \leq \gamma\epsilon_{opt}.$$

Substituting in the bounds of $\|Q^{\widehat{\mu},\widehat{\nu}} - \widehat{Q}^{\widehat{\mu},\widehat{\nu}}\|_\infty$, $\|Q^* - \widehat{Q}^{\mu^*,*}\|_\infty$, and $\|Q^* - \widehat{Q}^{*,\nu^*}\|_\infty$ in Lemma A.11, we arrive at the final bound for $\|Q^{\widehat{\mu},\widehat{\nu}} - Q^*\|_\infty$:

$$\|Q^{\widehat{\mu},\widehat{\nu}} - Q^*\|_\infty$$

$$\leq 4\gamma\left(\sqrt{\frac{c\log(c|\mathcal{S}||\mathcal{A}||\mathcal{B}|/[(1-\gamma)^2\delta])}{(1-\gamma)^3 N}} + \frac{c\log(c|\mathcal{S}||\mathcal{A}||\mathcal{B}|/[(1-\gamma)^2\delta])}{(1-\gamma)^2 N}\right) + \frac{4\gamma\epsilon_{opt}}{1-\gamma} + \gamma\epsilon_{opt}.$$

With a certain choice of $c$, we have $\|Q^{\widehat{\mu},\widehat{\nu}} - Q^*\|_\infty \leq 2\epsilon/3 + 5\gamma\epsilon_{opt}/(1-\gamma)$.

For the last argument in Theorem 3.2, by triangle inequality, with the same constant $c$ used above, we have

$$\|\widehat{Q}^{\widehat{\mu},\widehat{\nu}} - Q^*\|_\infty \leq \|Q^{\widehat{\mu},\widehat{\nu}} - Q^*\|_\infty + \|\widehat{Q}^{\widehat{\mu},\widehat{\nu}} - Q^{\widehat{\mu},\widehat{\nu}}\|_\infty \leq \epsilon + \frac{9\gamma\epsilon_{opt}}{1-\gamma},$$

which completes the proof. $\square$

## A.4 Proof of Corollary 3.3

We now prove Corollary 3.3, based on Theorem 3.2. For any state $s$, we have

$$V^*(s) - V^{\widetilde{\mu},*}(s) = \min_{\vartheta\in\Delta(\mathcal{B})} \mathbb{E}_{a\sim\mu^*(\cdot\,|\,s),b\sim\vartheta}\big[Q^*(s,a,b)\big] - \min_{\vartheta\in\Delta(\mathcal{B})} \mathbb{E}_{a\sim\widetilde{\mu}(\cdot\,|\,s),b\sim\vartheta}\big[Q^{\widetilde{\mu},*}(s,a,b)\big]$$

$$= \min_{\vartheta\in\Delta(\mathcal{B})} \mathbb{E}_{a\sim\mu^*(\cdot\,|\,s),b\sim\vartheta}\big[Q^*(s,a,b)\big] - \min_{\vartheta\in\Delta(\mathcal{B})} \mathbb{E}_{a\sim\widetilde{\mu}(\cdot\,|\,s),b\sim\vartheta}\big[Q^*(s,a,b)\big]$$

$$+ \min_{\vartheta\in\Delta(\mathcal{B})} \mathbb{E}_{a\sim\widetilde{\mu}(\cdot\,|\,s),b\sim\vartheta}\big[Q^*(s,a,b)\big] - \min_{\vartheta\in\Delta(\mathcal{B})} \mathbb{E}_{a\sim\widetilde{\mu}(\cdot\,|\,s),b\sim\vartheta}\big[Q^{\widetilde{\mu},*}(s,a,b)\big]$$

$$\leq \min_{\vartheta\in\Delta(\mathcal{B})} \mathbb{E}_{a\sim\mu^*(\cdot\,|\,s),b\sim\vartheta}\big[Q^*(s,a,b)\big] - \min_{\vartheta\in\Delta(\mathcal{B})} \mathbb{E}_{a\sim\widetilde{\mu}(\cdot\,|\,s),b\sim\vartheta}\big[Q^*(s,a,b)\big] + \gamma\|V^* - V^{\widetilde{\mu},*}\|_\infty$$

(A.44)

$$\leq \min_{\vartheta\in\Delta(\mathcal{B})} \mathbb{E}_{a\sim\mu^*(\cdot\,|\,s),b\sim\vartheta}\big[Q^*(s,a,b)\big] - \min_{\vartheta\in\Delta(\mathcal{B})} \mathbb{E}_{a\sim\mu^*(\cdot\,|\,s),b\sim\vartheta}\big[\widehat{Q}^{\widehat{\mu},\widehat{\nu}}(s,a,b)\big]$$

$$+ \min_{\vartheta\in\Delta(\mathcal{B})} \mathbb{E}_{a\sim\widetilde{\mu}(\cdot\,|\,s),b\sim\vartheta}\big[\widehat{Q}^{\widehat{\mu},\widehat{\nu}}(s,a,b)\big] - \min_{\vartheta\in\Delta(\mathcal{B})} \mathbb{E}_{a\sim\widetilde{\mu}(\cdot\,|\,s),b\sim\vartheta}\big[Q^*(s,a,b)\big]$$

$$+ \gamma\|V^* - V^{\widetilde{\mu},*}\|_\infty$$

(A.45)

$$\leq 2\|Q^* - \widehat{Q}^{\widehat{\mu},\widehat{\nu}}\|_\infty + \gamma\|V^* - V^{\widetilde{\mu},*}\|_\infty,$$

(A.46)

where (A.44) uses the fact that

$$\min_{\vartheta\in\Delta(\mathcal{B})} \mathbb{E}_{a\sim\widetilde{\mu}(\cdot\,|\,s),b\sim\vartheta}\big[Q^*(s,a,b)\big] - \min_{\vartheta\in\Delta(\mathcal{B})} \mathbb{E}_{a\sim\widetilde{\mu}(\cdot\,|\,s),b\sim\vartheta}\big[Q^{\widetilde{\mu},*}(s,a,b)\big]$$

$$\leq \max_{\vartheta\in\Delta(\mathcal{B})} \left| \mathbb{E}_{a\sim\widetilde{\mu}(\cdot\,|\,s),b\sim\vartheta}\big[Q^*(s,a,b)\big] - \mathbb{E}_{a\sim\widetilde{\mu}(\cdot\,|\,s),b\sim\vartheta}\big[Q^{\widetilde{\mu},*}(s,a,b)\big] \right| \leq \gamma\|V^* - V^{\widetilde{\mu},*}\|_\infty,$$

and (A.45) is due to the fact that

$$- \min_{\vartheta\in\Delta(\mathcal{B})} \mathbb{E}_{a\sim\mu^*(\cdot\,|\,s),b\sim\vartheta}\big[\widehat{Q}^{\widehat{\mu},\widehat{\nu}}(s,a,b)\big] + \min_{\vartheta\in\Delta(\mathcal{B})} \mathbb{E}_{a\sim\widetilde{\mu}(\cdot\,|\,s),b\sim\vartheta}\big[\widehat{Q}^{\widehat{\mu},\widehat{\nu}}(s,a,b)\big] \geq 0,$$

by definition of $\widetilde{\mu}$. Hence, (A.46), together with Theorem 3.2, implies that

$$V^* - V^{\widetilde{\mu},*} \leq \frac{2\|Q^* - \widehat{Q}^{\widehat{\mu},\widehat{\nu}}\|_\infty}{1-\gamma} = \widetilde{\epsilon}.$$

(A.47)

By similar arguments, we have

$$V^{*,\widetilde{\nu}} - V^* \leq \frac{2\|Q^* - \widehat{Q}^{\widehat{\mu},\widehat{\nu}}\|_\infty}{1-\gamma} = \widetilde{\epsilon}.$$

(A.48)

Combining (A.47) and (A.48) yields

$$V^{\widetilde{\mu},\widetilde{\nu}} - V^{\widetilde{\mu},*} \leq V^{*,\widetilde{\nu}} - V^{\widetilde{\mu},*} \leq 2\widetilde{\epsilon}, \qquad V^{*,\widetilde{\nu}} - V^{\widetilde{\mu},\widetilde{\nu}} \leq V^{*,\widetilde{\nu}} - V^{\widetilde{\mu},*} \leq 2\widetilde{\epsilon},$$

which completes the proof. $\qquad\square$

## A.5 Proof of Theorem 3.5

We now prove the second main result, Theorem 3.5. First, following the proof of Corollary 3.3, it suffices to prove that $V^* - V^{\widehat{\mu},*} \leq \widetilde{\epsilon}$, $V^{*,\widehat{\nu}} - V^* \leq \widetilde{\epsilon}$, since they together imply that $(\widehat{\mu},\widehat{\nu})$ is a $2\widetilde{\epsilon}$-Nash equilibrium. The following analysis is devoted to proving this argument.

The idea is similar to that presented in §A.3, i.e., we use the component-wise error decompositions in Lemma A.1, but use (A.4)-(A.5) instead. In particular, letting $\mu = \widehat{\mu}$ and $\nu = \widehat{\nu}$, we have

$$V^{\widehat{\mu},*} - V^* \geq -\|Q^{\widehat{\mu},*} - \widehat{Q}^{\widehat{\mu},*}\|_\infty - \epsilon_{opt} - \|\widehat{Q}^{\mu^*,*} - Q^*\|_\infty$$

(A.49)

$$V^{*,\widehat{\nu}} - V^* \leq \|Q^{*,\widehat{\nu}} - \widehat{Q}^{*,\widehat{\nu}}\|_\infty + \epsilon_{opt} + \|\widehat{Q}^{*,\nu^*} - Q^*\|_\infty.$$

(A.50)

Note that the bounds for $\|\widehat{Q}^{\mu^*,*} - Q^*\|_\infty$ and $\|\widehat{Q}^{*,\nu^*} - Q^*\|_\infty$ have already been established in Lemma A.11 (without dependence on $\epsilon_{opt}$ and the **Planning Oracle**). It now suffices to bound $\|Q^{\widehat{\mu},*} - \widehat{Q}^{\widehat{\mu},*}\|_\infty$ and $\|Q^{*,\widehat{\nu}} - \widehat{Q}^{*,\widehat{\nu}}\|_\infty$. For the former term, by Lemma A.1, we first have

$$\underbrace{\gamma(I - \gamma\widehat{P}^{\widehat{\mu},\nu(\widehat{\mu})})^{-1}(P - \widehat{P})V^{\widehat{\mu},\nu(\widehat{\mu})}}_{Q^{\widehat{\mu},*} - \widehat{Q}^{\widehat{\mu},\nu(\widehat{\mu})}} \leq Q^{\widehat{\mu},*} - \widehat{Q}^{\widehat{\mu},*} \leq \underbrace{\gamma(I - \gamma P^{\widehat{\mu},\widehat{\nu(\widehat{\mu})}})^{-1}(P - \widehat{P})\widehat{V}^{\widehat{\mu},\widehat{\nu(\widehat{\mu})}}}_{Q^{\widehat{\mu},\nu(\widehat{\mu})} - \widehat{Q}^{\widehat{\mu},*}}.$$

Thus, we know that

$$\left\|Q^{\widehat{\mu},*} - \widehat{Q}^{\widehat{\mu},*}\right\|_\infty \tag{A.51}$$
$$\leq \max\left\{\gamma\left\|(I - \gamma P^{\widehat{\mu},\widehat{\nu(\widehat{\mu})}})^{-1}(P - \widehat{P})\widehat{V}^{\widehat{\mu},\widehat{\nu(\widehat{\mu})}}\right\|_\infty, \; \gamma\left\|(I - \gamma \widehat{P}^{\widehat{\mu},\nu(\widehat{\mu})})^{-1}(P - \widehat{P})V^{\widehat{\mu},\nu(\widehat{\mu})}\right\|_\infty\right\}.$$

The first term in the $\max$ operator, where the policies in the pair $(\widehat{\mu}, \widehat{\nu(\widehat{\mu})})$ are both obtained from the empirical model $\widehat{\mathcal{G}}$, can be bounded similarly as that for $\|Q^{\widehat{\mu},\widehat{\nu}} - \widehat{Q}^{\widehat{\mu},\widehat{\nu}}\|_\infty$ in Lemma A.11. Specifically, following (A.33)-(A.35), we have

$$\gamma\left\|(I - \gamma P^{\widehat{\mu},\widehat{\nu(\widehat{\mu})}})^{-1}(P - \widehat{P})\widehat{V}^{\widehat{\mu},*}\right\|_\infty$$
$$\leq \gamma\left\|(I - \gamma P^{\widehat{\mu},\widehat{\nu(\widehat{\mu})}})^{-1}(P - \widehat{P})\widehat{V}^*\right\|_\infty + \gamma\left\|(I - \gamma P^{\widehat{\mu},\widehat{\nu(\widehat{\mu})}})^{-1}(P - \widehat{P})(\widehat{V}^{\widehat{\mu},*} - \widehat{V}^*)\right\|_\infty \tag{A.52}$$

$$\leq \gamma\left\|(I - \gamma P^{\widehat{\mu},\widehat{\nu(\widehat{\mu})}})^{-1}\left|(P - \widehat{P})\widehat{V}^*\right|\right\|_\infty + \frac{2\gamma\epsilon_{opt}}{1 - \gamma}, \tag{A.53}$$

where (A.52) uses the triangle inequality, and (A.53) is due to the optimization error of $\widehat{\mu}$. Then, to bound $\gamma\left\|(I - \gamma P^{\widehat{\mu},\widehat{\nu(\widehat{\mu})}})^{-1}\left|(P - \widehat{P})\widehat{V}^*\right|\right\|_\infty$, the rest of the proof is analogous to the derivations in (A.36)-(A.38), by replacing $\widehat{\nu}$ therein by $\widehat{\nu(\widehat{\mu})}$, and bound $\|\widehat{V}^{\widehat{\mu},*} - \widehat{V}^*\|_\infty$ by $\epsilon_{opt}$. Solving for $\|Q^{\widehat{\mu},\widehat{\nu(\widehat{\mu})}} - \widehat{Q}^{\widehat{\mu},*}\|_\infty$ yields the desired bound for the first term in the $\max$ in (A.51), namely, there exists some constant $c$ such that with probability greater than $1 - \delta$,

$$\left\|Q^{\widehat{\mu},\widehat{\nu(\widehat{\mu})}} - \widehat{Q}^{\widehat{\mu},*}\right\|_\infty$$
$$\leq \frac{\gamma}{1 - \alpha'_{\delta,N}}\left(\sqrt{\frac{c\log(c(C + 1)|\mathcal{S}||\mathcal{A}||\mathcal{B}|/[(1 - \gamma)^4\delta])}{(1 - \gamma)^3 N}} + \frac{c\log(c(C + 1)|\mathcal{S}||\mathcal{A}||\mathcal{B}|/[(1 - \gamma)^4\delta])}{(1 - \gamma)^2 N}\right)$$
$$+ \frac{1}{1 - \alpha'_{\delta,N}} \cdot \frac{\gamma\epsilon_{opt}}{(1 - \gamma)}\left(1 + \sqrt{\frac{\log(c(C + 1)|\mathcal{S}||\mathcal{A}||\mathcal{B}|/[(1 - \gamma)^4\delta])}{N}}\right), \tag{A.54}$$

where $\alpha'_{\delta,N}$ is defined as

$$\alpha'_{\delta,N} = \frac{\gamma}{1 - \gamma}\sqrt{\frac{2\log(8(C + 1)|\mathcal{S}||\mathcal{A}||\mathcal{B}|/[(1 - \gamma)^4\delta])}{N}}.$$

For the second term in the $\max$ in (A.51), note that $\widehat{\mu}$ is obtained from $\widehat{\mathcal{G}}$, while $\nu(\widehat{\mu})$ is obtained from the true model $\mathcal{G}$. By Lemma A.10, it holds that

$$\gamma\left\|(I - \gamma \widehat{P}^{\widehat{\mu},\nu(\widehat{\mu})})^{-1}\left|(P - \widehat{P})V^{\widehat{\mu},*}\right|\right\|_\infty$$
$$\leq \gamma\sqrt{\frac{2\log\left(8(C + 1)|\mathcal{S}||\mathcal{A}||\mathcal{B}|/[(1 - \gamma)^4\delta]\right)}{N}}\left\|(I - \gamma \widehat{P}^{\widehat{\mu},\nu(\widehat{\mu})})^{-1}\sqrt{\mathrm{Var}_P(V^{\widehat{\mu},*})}\right\|_\infty + \frac{\gamma\Delta'_{\delta,N}}{1 - \gamma}$$
$$\leq \gamma\sqrt{\frac{2\log\left(8(C + 1)|\mathcal{S}||\mathcal{A}||\mathcal{B}|/[(1 - \gamma)^4\delta]\right)}{N}} \cdot \left[\left\|(I - \gamma \widehat{P}^{\widehat{\mu},\nu(\widehat{\mu})})^{-1}\left(\sqrt{\mathrm{Var}_{\widehat{P}}(\widehat{V}^{\widehat{\mu},\nu(\widehat{\mu})})}\right.\right.\right.$$
$$\left.+ \sqrt{\mathrm{Var}_{\widehat{P}}(V^{\widehat{\mu},*} - \widehat{V}^{\widehat{\mu},\nu(\widehat{\mu})})}\right)\Big\|_\infty + \left\|(I - \gamma \widehat{P}^{\widehat{\mu},\nu(\widehat{\mu})})^{-1}\right|\sqrt{\mathrm{Var}_P(V^{\widehat{\mu},*})}$$
$$\left.- \sqrt{\mathrm{Var}_{\widehat{P}}(V^{\widehat{\mu},*})}\Big|\right\|_\infty\Big] + \frac{\gamma\Delta'_{\delta,N}}{1 - \gamma} \tag{A.55}$$
$$\leq \gamma\sqrt{\frac{2\log\left(8(C + 1)|\mathcal{S}||\mathcal{A}||\mathcal{B}|/[(1 - \gamma)^4\delta]\right)}{N}}\left(\sqrt{\frac{2}{(1 - \gamma)^3}} + \frac{\|Q^{\widehat{\mu},*} - \widehat{Q}^{\widehat{\mu},\nu(\widehat{\mu})}\|_\infty}{1 - \gamma}\right) + \frac{\gamma\Delta'_{\delta,N}}{1 - \gamma} \tag{A.56}$$
$$+ \frac{\gamma}{1 - \gamma}\sqrt{\frac{2\log\left(8(C + 1)|\mathcal{S}||\mathcal{A}||\mathcal{B}|/[(1 - \gamma)^4\delta]\right)}{N}}\left\|\left|\sqrt{\mathrm{Var}_P(V^{\widehat{\mu},*})} - \sqrt{\mathrm{Var}_{\widehat{P}}(V^{\widehat{\mu},*})}\right|\right\|_\infty,$$

where (A.55) uses the norm-like triangle-inequality property of $\sqrt{\mathrm{Var}_P(V)}$ and triangle inequality, (A.56) is due to Lemma A.3, and the facts that $\sqrt{\mathrm{Var}_P(X)} \leq \|X\|_\infty$, $\|V^{\widehat{\mu},*} - \widehat{V}^{\widehat{\mu},\nu(\widehat{\mu})}\|_\infty \leq \|Q^{\widehat{\mu},*} - \widehat{Q}^{\widehat{\mu},\nu(\widehat{\mu})}\|_\infty$, and Lemma A.2. Moreover, notice that

$$\left\| \left| \sqrt{\mathrm{Var}_P(V^{\widehat{\mu},*})} - \sqrt{\mathrm{Var}_{\widehat{P}}(V^{\widehat{\mu},*})} \right| \right\|_\infty$$

$$\leq \left\| \left| \sqrt{\mathrm{Var}_P(V^{\widehat{\mu},*})} - \sqrt{\mathrm{Var}_P(V^*)} \right| \right\|_\infty + \left\| \left| \sqrt{\mathrm{Var}_{\widehat{P}}(V^{\widehat{\mu},*})} - \sqrt{\mathrm{Var}_{\widehat{P}}(V^*)} \right| \right\|_\infty$$

$$+ \left\| \left| \sqrt{\mathrm{Var}_P(V^*)} - \sqrt{\mathrm{Var}_{\widehat{P}}(V^*)} \right| \right\|_\infty \tag{A.57}$$

$$\leq \left\| \sqrt{\mathrm{Var}_P(V^{\widehat{\mu},*} - V^*)} \right\|_\infty + \left\| \sqrt{\mathrm{Var}_{\widehat{P}}(V^{\widehat{\mu},*} - V^*)} \right\|_\infty + \left\| \sqrt{\left| \mathrm{Var}_P(V^*) - \mathrm{Var}_{\widehat{P}}(V^*) \right|} \right\|_\infty \tag{A.58}$$

$$\leq 2 \|V^{\widehat{\mu},*} - V^*\|_\infty + \sqrt{\left\| \left| \mathrm{Var}_P(V^*) - \mathrm{Var}_{\widehat{P}}(V^*) \right| \right\|_\infty}, \tag{A.59}$$

where (A.57) uses triangle inequality, (A.58) uses the norm-like triangle inequality of $\sqrt{\mathrm{Var}_P(V)}$ and $\sqrt{\mathrm{Var}_{\widehat{P}}(V)}$, and the fact $|\sqrt{X} - \sqrt{Y}| \leq \sqrt{|X - Y|}$ for $X, Y \geq 0$, and (A.59) uses $\sqrt{\mathrm{Var}_P(X)} \leq \|X\|_\infty$ and the definition of $\|\cdot\|_\infty$. In addition, we know that with probability at least $1 - \delta$,

$$\left\| \mathrm{Var}_P(V^*) - \mathrm{Var}_{\widehat{P}}(V^*) \right\|_\infty = \left\| (P - \widehat{P})(V^*)^2 - \left( (PV^*)^2 - (\widehat{P}V^*)^2 \right) \right\|_\infty$$

$$\leq \left\| (P - \widehat{P})(V^*)^2 \right\|_\infty + \left\| (PV^*)^2 - (\widehat{P}V^*)^2 \right\|_\infty$$

$$\leq \frac{1}{(1-\gamma)^2} \sqrt{\frac{2\log(2|\mathcal{S}||\mathcal{A}||\mathcal{B}|/\delta)}{N}} + \frac{2}{1-\gamma} \|(P - \widehat{P})V^*\|_\infty$$

$$\leq \frac{3}{(1-\gamma)^2} \sqrt{\frac{2\log(2|\mathcal{S}||\mathcal{A}||\mathcal{B}|/\delta)}{N}}, \tag{A.60}$$

due to Hoeffding bound and $\|V^*\|_\infty \leq 1/(1-\gamma)$.

Combining (A.56), (A.59), and (A.60) yields

$$\left\| Q^{\widehat{\mu},*} - \widehat{Q}^{\widehat{\mu},\nu(\widehat{\mu})} \right\|_\infty$$

$$\leq \gamma \sqrt{\frac{2\log\left(8(C+1)|\mathcal{S}||\mathcal{A}||\mathcal{B}|/[(1-\gamma)^4\delta]\right)}{N}} \left( \sqrt{\frac{2}{(1-\gamma)^3}} + \frac{\|Q^{\widehat{\mu},*} - \widehat{Q}^{\widehat{\mu},\nu(\widehat{\mu})}\|_\infty}{1-\gamma} \right) + \frac{\gamma\Delta'_{\delta,N}}{1-\gamma}$$

$$+ \frac{\gamma}{1-\gamma} \sqrt{\frac{2\log\left(8(C+1)|\mathcal{S}||\mathcal{A}||\mathcal{B}|/[(1-\gamma)^4\delta]\right)}{N}} \left( 2\|V^{\widehat{\mu},*} - V^*\|_\infty \right.$$

$$\left. + \sqrt{\frac{3}{(1-\gamma)^2} \sqrt{\frac{2\log(2|\mathcal{S}||\mathcal{A}||\mathcal{B}|/\delta)}{N}}} \right).$$

Solving for $\|Q^{\widehat{\mu},*} - \widehat{Q}^{\widehat{\mu},\nu(\widehat{\mu})}\|_\infty$ further leads to

$$\left\| Q^{\widehat{\mu},*} - \widehat{Q}^{\widehat{\mu},\nu(\widehat{\mu})} \right\|_\infty$$

$$\leq \frac{\gamma}{1-\alpha'_{\delta,N}} \left( \sqrt{\frac{c\log(c(C+1)|\mathcal{S}||\mathcal{A}||\mathcal{B}|/[(1-\gamma)^4\delta])}{(1-\gamma)^3 N}} + \frac{c\log(c(C+1)|\mathcal{S}||\mathcal{A}||\mathcal{B}|/[(1-\gamma)^4\delta])}{(1-\gamma)^2 N} \right)$$

$$+ \frac{1}{1-\alpha'_{\delta,N}} \cdot \frac{\gamma}{1-\gamma} \sqrt{\frac{2\log\left(8(C+1)|\mathcal{S}||\mathcal{A}||\mathcal{B}|/[(1-\gamma)^4\delta]\right)}{N}} \left( 2\|V^{\widehat{\mu},*} - V^*\|_\infty \right.$$

$$\left. + \frac{1}{1-\gamma} \sqrt[4]{\frac{c\log(c(C+1)|\mathcal{S}||\mathcal{A}||\mathcal{B}|/\delta)}{N}} \right), \tag{A.61}$$

for some absolute constant $c$.

Now we substitute (A.54) and (A.61) into (A.51), to complete the bound in (A.49). If the first term in the max in (A.51) is larger, and noticing that the choice of $N$ in the theorem can make $\alpha'_{\delta,N} < 1/5$, (A.49), (A.51), (A.54), and Lemma A.11 together lead to

$$V^* - V^{\widehat{\mu},*}$$

$$\leq \frac{5\gamma}{2}\left(\sqrt{\frac{c\log(c(C+1)|\mathcal{S}||\mathcal{A}||\mathcal{B}|/[(1-\gamma)^4\delta])}{(1-\gamma)^3N}} + \frac{c\log(c(C+1)|\mathcal{S}||\mathcal{A}||\mathcal{B}|/[(1-\gamma)^4\delta])}{(1-\gamma)^2N}\right)$$

$$+ \frac{5\gamma\epsilon_{opt}}{2(1-\gamma)} + \epsilon_{opt}, \tag{A.62}$$

with some absolute constant $c$, where we have replaced the term $\log(1/(1-\gamma)^2)$ in the bounds for $\|Q^* - \widehat{Q}^{\mu^*,*}\|_\infty$ and $\|Q^* - \widehat{Q}^{*,\nu^*}\|_\infty$ in Lemma A.11 (including that in the definition of $\alpha_{\delta,N}$) by $\log((C+1)/(1-\gamma)^4)$, a larger number. If the second term in the max in (A.51) is larger, (A.49), (A.51), (A.61), and Lemma A.11 together yield

$$V^* - V^{\widehat{\mu},*}$$

$$\leq \frac{5\gamma}{2}\left(\sqrt{\frac{c\log(c(C+1)|\mathcal{S}||\mathcal{A}||\mathcal{B}|/[(1-\gamma)^4\delta])}{(1-\gamma)^3N}} + \frac{c\log(c(C+1)|\mathcal{S}||\mathcal{A}||\mathcal{B}|/[(1-\gamma)^4\delta])}{(1-\gamma)^2N}\right)$$

$$+ \frac{5}{4}\cdot\frac{\gamma}{1-\gamma}\sqrt{\frac{2\log\left(8(C+1)|\mathcal{S}||\mathcal{A}||\mathcal{B}|/[(1-\gamma)^4\delta]\right)}{N}}\left(2\|V^{\widehat{\mu},*} - V^*\|_\infty\right.$$

$$\left.+ \frac{1}{1-\gamma}\sqrt[4]{\frac{c\log(c(C+1)|\mathcal{S}||\mathcal{A}||\mathcal{B}|/\delta)}{N}}\right) + \epsilon_{opt},$$

where we have used the fact that $\alpha'_{\delta,N} < 1/5$. Taking infinity norm on both sides and solving for $\|V^{\widehat{\mu},*} - V^*\|_\infty$, we have

$$V^* - V^{\widehat{\mu},*} \leq \|V^{\widehat{\mu},*} - V^*\|_\infty \leq 5\gamma\left(\sqrt{\frac{c\log(c(C+1)|\mathcal{S}||\mathcal{A}||\mathcal{B}|/[(1-\gamma)^4\delta])}{(1-\gamma)^3N}} + \right. \tag{A.63}$$

$$\left.\frac{c\log(c(C+1)|\mathcal{S}||\mathcal{A}||\mathcal{B}|/[(1-\gamma)^4\delta])}{(1-\gamma)^2N}\right) + \frac{5\gamma}{2(1-\gamma)^2}\left(\frac{c\log(c(C+1)|\mathcal{S}||\mathcal{A}||\mathcal{B}|/\delta)}{N}\right)^{3/4} + 2\epsilon_{opt},$$

with some absolute constant $c$ (which can be different from that in (A.62)). Using the choice of $N$ in the theorem, and combining (A.62) and (A.63), we finally have $V^* - V^{\widehat{\mu},*} \leq \epsilon + 4\epsilon_{opt}/(1-\gamma)$. Note that on the right-hand of (A.63), the $N$ that makes the third term to be $\mathcal{O}(\epsilon)$ is $\widetilde{\mathcal{O}}(1/[(1-\gamma)^{8/3}\epsilon^{4/3}])$, which is dominated by $\widetilde{\mathcal{O}}(1/[(1-\gamma)^3\epsilon^2])$ when $\epsilon \in (0, 1/(1-\gamma)^{1/2}]$. In addition, to make $\alpha'_{\delta,N} < 1/5$, $N$ should be larger than $\mathcal{O}(1/(1-\gamma)^2)$, this is consistent with both the first and third terms on the right-hand of (A.63) to be $\mathcal{O}(1/(1-\gamma)^{1/2})$, determining the allowed range of $\epsilon$ to be $(0, 1/(1-\gamma)^{1/2}]$. This proves the first bound in the theorem.

The proof for completing the bound in (A.50) is analogous: using Lemmas A.10 and A.1 to bound $\|Q^{*,\widehat{\nu}} - \widehat{Q}^{*,\widehat{\nu}}\|_\infty$, which is then substituted into (A.50). This completes the proof. □

# B  On the Lower Bound

Now we discuss the lower bound of the sample complexity given in Lemma 3.1.

**Proof of Lemma 3.1.**  The proof follows by recalling the hard cases of MDPs considered in [26] or [47], and replacing each action $a$ therein by a joint-action $(a, b)$. Without loss of generality, suppose $|\mathcal{A}| \geq |\mathcal{B}|$. Then, we design a Markov game such that agent 2 has no effect on either the reward or the transition. Thus, finding an NE is now the same as agent 1 finding the optimal value/policy. By the arguments in [26, 47], the sample complexity is at least $\Omega\left(|\mathcal{S}| \cdot \max\{|\mathcal{A}|, |\mathcal{B}|\} \cdot (1-\gamma)^{-3}\epsilon^{-2}\right)$. Notice that $\max\{|\mathcal{A}|, |\mathcal{B}|\} = (|\mathcal{A}| + |\mathcal{B}| + ||\mathcal{A}| - |\mathcal{B}||)/2$, we obtain the lower bound. □

**Challenge in Obtaining** $\Omega(|\mathcal{A}||\mathcal{B}|)$. Though the proof above seems straightforward, and the result can be obtained in several different ways (either as above or another treatment of turn-based MGs or the approaches to be introduced), we highlight the challenge in obtaining a tighter lower bound of order $\Omega(|\mathcal{A}||\mathcal{B}|)$ (not $\Omega(|\mathcal{A}| + |\mathcal{B}|)$), if one follows the lower bound proof framework before [26, 47]. The core idea in those proofs is to create a class of $\mathcal{O}(|\mathcal{S}||\mathcal{A}|)$ number of MDPs, which are hard to distinguish from each other. More specifically, there is a null case MDP in the hypothesis testing, and every other MDP in the class, as alternative cases, corresponds to every $(s, a)$ pair. Each alternative case is generated by *one single* change of the transition probability at this $(s, a)$, while leading to a large enough difference of the Q-value from the null case. Thus, the optimal action at this alternative case is *changed* from the original one in the null case to this $a$. Then, it can be shown that for any algorithm, if it correctly outputs the Q-value in the alternative case with high probability, then it must have sampled $\Omega((1 - \gamma)^{-3}\epsilon^{-2})$ samples at this $(s, a)$ pair in the null case. As this holds for all $\mathcal{O}(|\mathcal{S}||\mathcal{A}|)$ alternative cases, and they have no overlapped changes of $(s, a)$ pairs from each other, leading to a total number of $\Omega(|\mathcal{S}||\mathcal{A}|(1 - \gamma)^{-3}\epsilon^{-2})$ samples.

In contrast, in zero-sum MGs, at each state $s$, a zero-sum matrix game is solved. Following the similar idea, one does need to construct $\mathcal{O}(|\mathcal{S}||\mathcal{A}||\mathcal{B}|)$ alternative cases, by only making $\mathcal{O}(1)$ number of changes in the Q-value in each of them, so that the Nash equilibrium Q-value at each $s$ is changed by a relatively large amount. However, this seems challenging to achieve in general, as the NE value of zero-sum matrix games is *not sensitive* to the small number of element changes in the payoff matrices. It is possible to change the NE value by a relatively large amount by changing one row/column of the payoff matrix, i.e., $\mathcal{O}(|\mathcal{A}|)$ (or $\mathcal{O}(|\mathcal{B}|)$) number of changes. Nonetheless, this will only give us essentially $\mathcal{O}(|\mathcal{B}|)$ or $\mathcal{O}(|\mathcal{A}|)$ hard alternative cases, which ends up with the same $\Omega(|\mathcal{A}| + |\mathcal{B}|)$ result as Lemma 3.1. In fact, if the null case admits a pure NE at some state $s$, it suffices to use samples to accurately estimate the payoff elements in the row and column that this pure NE point occupies, while all other payoff values do not need to be accurately estimated. This ends up with a $\Omega(|\mathcal{A}| + |\mathcal{B}|)$ lower bound, too. For more general cases with mixed NE, the change of the NE value is still small, with only $\mathcal{O}(1)$ elements changed with a small magnitude. This can be evidenced either by the stability of the NE in this case against the payoff perturbation [67], or by the sensitivity analysis of the equivalent linear program of the game [68] against the problem data [69]. Hence, the existing technique based on constructing state-by-state bandit/matrix game may not be sufficient. More sophisticated coupling among the matrix games at different states may be required to establish harder MG cases.

On the other hand, interestingly, we note that there are some results on the *payoff query complexity*, i.e., the number of queries for the elements in the payoff matrix, for finding the NE [70, 71]. It is possible to use $\mathcal{O}(k \log(k)/\epsilon^2)$ queries to find the $\epsilon$-NE in zero-sum matrix games when $|\mathcal{A}| = |\mathcal{B}|$, where $k = |\mathcal{A}| = |\mathcal{B}|$ [71]. Note that the lower bound given in [71], though being $\Omega(k^2)$, requires the accuracy $\epsilon \leq 1/k$ to be small, which cannot be used in our previous analysis with a dimension-free choice of $\epsilon$. From a different angle, these results imply that it may indeed be unnecessary to accurately estimate *all* elements in the matrix, in order to obtain an approximate Nash equilibrium.

In light of these observations, we conjecture that the lower bound of $\Omega(|\mathcal{A}| + |\mathcal{B}|)$ is indeed unimprovable, which can be matched by some other (possibly model-free) MARL algorithms. Interestingly, such a $\Omega(|\mathcal{A}| + |\mathcal{B}|)$ lower bound on *regret* has been provided recently in [36], though in a different MARL setting. This $\Omega(|\mathcal{A}| + |\mathcal{B}|)$ lower bound was also shown to be attained by no-regret learning algorithms there, when horizon $H = 1$. The challenge of matching the lower bound for actual multi-step Markov games was also acknowledged there.

## C  A Smooth Planning Oracle

We now show that solving the regularized matrix game induced by $\widehat{Q}^*$, see (3.2), leads to a smooth **Planning Oracle** (see Definition 3.4).

**Lemma C.1.** Suppose that the regularizers $\Omega_i$ for $i = 1, 2$ in (3.2) are twice continuously differentiable and strongly convex. Suppose that the solution policy pair $(\widehat{\mu}(\cdot \mid s), \widehat{\nu}(\cdot \mid s))$ of (3.2) lies in the relative interior of the simplexes $\Delta(\mathcal{A})$ and $\Delta(\mathcal{B})$, respectively. Then, $(\widehat{\mu}, \widehat{\nu})$ is smooth with respect to $\widehat{Q}^*$, namely, this **Planning Oracle** follows Definition 3.4 with some constant $C$.

*Proof.* Let $Q_s := \widehat{Q}^*(s, \cdot, \cdot) \in \mathbb{R}^{|\mathcal{A}| \times |\mathcal{B}|}$ denote the payoff matrix of the game at state $s$. Note that $Q_s \in [0, (1 - \gamma)^{-1}]^{|\mathcal{A}| \times |\mathcal{B}|}$, $u \in [0, 1]^{|\mathcal{A}|}$ and $\vartheta \in [0, 1]^{|\mathcal{B}|}$. First, as the solution to (3.2) lies in the

relative interior of the simplex, by first-order optimality, we have that for each $s \in \mathcal{S}$

$$\nabla_u \Omega_1(u) - Q_s \vartheta = 0, \qquad \nabla_\vartheta \Omega_2(\vartheta) + Q_s^\top u = 0, \qquad (\text{C.1})$$

whose solution is unique. Define a function $F : \mathbb{R}^{|\mathcal{A}|} \times \mathbb{R}^{|\mathcal{B}|} \times \mathbb{R}^{|\mathcal{A}||\mathcal{B}|} \to \mathbb{R}^{|\mathcal{A}|+|\mathcal{B}|}$ as follows, such that (C.1) is equivalent to

$$F\big(u, \vartheta, \mathrm{vec}(Q_s)\big) := \begin{bmatrix} \nabla_u \Omega_1(u) - Q_s \vartheta \\ \nabla_\vartheta \Omega_2(\vartheta) + Q_s^\top u \end{bmatrix} = 0.$$

As the solution to (C.1) lies in the relative interior of $\Delta(\mathcal{A}) \times \Delta(\mathcal{B})$, for any choice of $Q_s \in \mathbb{R}^{|\mathcal{A}| \times |\mathcal{B}|}$ (not just $[0, (1-\gamma)^{-1}]^{|\mathcal{A}| \times |\mathcal{B}|}$), the domain of $F$ can be specified as $\Delta^o(\mathcal{A}) \times \Delta^o(\mathcal{B}) \times \Lambda$, where $\Delta^o(\mathcal{A})$ and $\Delta^o(\mathcal{B})$ denote the interiors of $\Delta(\mathcal{A})$ and $\Delta(\mathcal{B})$, respectively, and $\Lambda \subset \mathbb{R}^{|\mathcal{A}||\mathcal{B}|}$ denotes some open set that contains $[0, (1-\gamma)^{-1}]^{|\mathcal{A}||\mathcal{B}|}$.

Notice that the Jacobian of $F$ with respect to $[u^\top \ \vartheta^\top]^\top$ is

$$M\big(u, \vartheta, \mathrm{vec}(Q_s)\big) := \begin{bmatrix} \frac{\partial F}{\partial u} & \frac{\partial F}{\partial \vartheta} \end{bmatrix} = \begin{bmatrix} \nabla_u^2 \Omega_1(u) & -Q_s \\ Q_s^\top & \nabla_\vartheta^2 \Omega_2(\vartheta) \end{bmatrix}, \qquad (\text{C.2})$$

which is always invertible for any point in $\Delta^o(\mathcal{A}) \times \Delta^o(\mathcal{B}) \times \Lambda$. This is because $\Omega_i$ are strongly convex, and thus the real parts of the eigenvalues of the matrix, which are the eigenvalues of $(M + M^\top)/2$, are always positive and uniformly lower bounded, namely, there exists some constant $\eta > 0$, such that

$$\min_i \lambda_i \big( M(u, \vartheta, \mathrm{vec}(Q_s)) + M^\top(u, \vartheta, \mathrm{vec}(Q_s)) \big) \geq 2\eta > 0,$$

with $\lambda_i(\cdot)$ being the eigenvalues of the corresponding matrix. This further implies that for any $(u, \vartheta, \mathrm{vec}(Q_s)) \in \Delta^o(\mathcal{A}) \times \Delta^o(\mathcal{B}) \times \Lambda$,

$$\big\| M(u, \vartheta, \mathrm{vec}(Q_s))^{-1} \big\|_2 = \frac{1}{\min_i \ \sigma_i(M(u, \vartheta, \mathrm{vec}(Q_s)))}$$

$$\leq \frac{2}{\min_i \ \lambda_i(M(u, \vartheta, \mathrm{vec}(Q_s)) + M^\top(u, \vartheta, \mathrm{vec}(Q_s)))} \leq \frac{1}{\eta},$$

where $\sigma_i$ is the singular value of $M$.

By the implicit function theorem [72], for any point that solves $F(u, \vartheta, Q_s) = 0$, since $M(u, \vartheta, \mathrm{vec}(Q_s))$ is invertible, there exists a neighborhood $U \subseteq \Delta^o(\mathcal{A})$, $V \subseteq \Delta^o(\mathcal{B})$, and $W \subseteq \Lambda$ around it, such that $[u^\top \ \vartheta^\top]^\top \in U \times V$ is a unique function of $\mathrm{vec}(Q_s)$ for all $\mathrm{vec}(Q_s) \in W$, and

$$\frac{\partial [u^\top v^\top]^\top}{\partial \mathrm{vec}(Q_s)} = -\begin{bmatrix} \frac{\partial F}{\partial u} & \frac{\partial F}{\partial \vartheta} \end{bmatrix}^{-1} \cdot \frac{\partial F}{\partial \mathrm{vec}(Q_s)} = -M(u, \vartheta, \mathrm{vec}(Q_s))^{-1} \cdot \begin{bmatrix} \overline{\phantom{xx}} -\vartheta^\top \otimes e_1 \overline{\phantom{xx}} \\ \vdots \\ \overline{\phantom{xx}} -\vartheta^\top \otimes e_{|\mathcal{A}|} \overline{\phantom{xx}} \\ u^\top \quad 0 \quad \cdots \quad 0 \\ \ddots \\ 0 \quad 0 \quad \cdots \quad u^\top \end{bmatrix},$$

where $e_i \in \mathbb{R}^{|\mathcal{B}|}$ is an all-zero vector except that the $i$-th element is 1. Thus, we have

$$\left\| \frac{\partial [u^\top v^\top]^\top}{\partial \mathrm{vec}(Q_s)} \right\|_2 \leq \big\| M(u, \vartheta, \mathrm{vec}(Q_s))^{-1} \big\|_2 \cdot \left\| \frac{\partial F}{\partial \mathrm{vec}(Q_s)} \right\|_2 \leq \frac{|\mathcal{A}||\mathcal{B}|}{\eta} \cdot \left\| \frac{\partial F}{\partial \mathrm{vec}(Q_s)} \right\|_1 = \frac{|\mathcal{A}||\mathcal{B}|}{\eta}.$$

Notice that this is a uniform bound on the gradient of the implicit function, at any point in $\Delta^o(\mathcal{A}) \times \Delta^o(\mathcal{B}) \times \Lambda$, which together with the mean value theorem leads to

$$\big\| [u_1^\top v_1^\top] - [u_2^\top v_2^\top] \big\|_2 \leq \frac{|\mathcal{A}||\mathcal{B}|}{\eta} \cdot \big\| \mathrm{vec}(Q_{s,1}) - \mathrm{vec}(Q_{s,2}) \big\|_2,$$

where the pair $(u_i, \vartheta_i)$ is the unique solution of $F = 0$ corresponding to $Q_{s,i}$. By the equivalence of norms and considering all $s \in \mathcal{S}$, we can find some constant $C$ (which may depend on $|\mathcal{A}|$ and $|\mathcal{B}|$ polynomially) as the smooth coefficient, and this completes the proof. $\qquad \square$

To ensure that the solution $(\widehat{\mu}(\cdot \mid s), \widehat{\nu}(\cdot \mid s))$ of (3.2) lies in the relative interior of the simplexes, the common choice of *steep* regularizers will suffice [61]. The steep regularizer means that for any $u$ (resp. $\vartheta$) on the boundary of the simplex $\Delta(\mathcal{A})$ (resp. $\Delta(\mathcal{B})$), and for every interior sequence $u_n \to u$ (resp. $\vartheta_n \to \vartheta$) that approaches it, it holds that $\left\| \frac{d\Omega_1(u)}{du} \big|_{u=u_n} \right\|_2 \to \infty$ (resp. $\left\| \frac{d\Omega_2(\vartheta)}{d\vartheta} \big|_{\vartheta=\vartheta_n} \right\|_2 \to \infty$). This way, the optimizer cannot occur on the boundary of the simplexes. Examples of steep regularizers in Lemma C.1 include the commonly used negative entropy, Tsallis entropy and Rényi entropy with certain parameters; see [61] for more details.