[Reviews · NeurIPS 2020]

Review 1

Summary and Contributions: The rebuttal and discussion confirms my positive impression of this article. --- The title accurately summarizes the article. It addresses a zero-sum Markov game setting in which a generative model is available, and establishes sample complexity bounds for model-based RL to find epsilon-Nash Equilibrium solutions. Additional derivations of corresponding lower bounds establish the near optimality of the achieved sample complexity.

Strengths: The obtained results are derived from general assumptions, and thus widely applicable. The work is fundamental and theoretic in nature, contributing to the frontier of theoretical guarantees in multi-agent reinforcement learning, which is highly relevant to the NeurIPS community. The article conveys an astounding technical depth while maintaining a coherent organisation and surprisingly good readability.

Weaknesses: The very fundamental nature is a limitation of this work in terms of the audience that it may find. The broader impact section could be used to reflect on the impact of complexity bounds on applications. As-is it rather merely re-iterates the contributions.

Correctness: The arguments in the main text are delivered meticulously and appear consistent. To the degree that I sampled from the appendix it seems correct.

Clarity: The presentation is outstanding. Clean use of technical concepts, and reference to their mathematical nature (such as the "saddle-point problem"), ease the reading of this technically dense material. The article makes a good compromise of presenting a roadmap of proofs in the main body, and deferring technical detail to the appendix.

Relation to Prior Work: Demarcation to prior work is detailed and precise.

Reproducibility: Yes

Additional Feedback: The total variation distance could be mentioned in the appendix for completeness sake. - "(3.3) is built upon Theorem 3.2" could better read "Corrolary 3.3 ..."


Review 2

Summary and Contributions: The paper provides an approach to learn a zero-sum Markov game from a sampling model and shows how much sampling is needed to achieve eps-optimal values and policies.

Strengths: It builds on recent results in the single agent setting and contributes careful formal comparisons and analyses to the question.

Weaknesses: The primary conceptual idea here is the smooth planning oracle. I think the paper would be improved if this idea were applied to/demonstrated in some simple games to build the reader's intuition.

Correctness: The main proofs are in the supplementary material, so it's hard to tell from the paper if the results are correct. Nevertheless, the results are presented carefully and connected to closely related results and I don't see any obvious flaws.

Clarity: Clarity is well above the bar.

Relation to Prior Work: The extensive bibliography helps situate the work in terms of earlier efforts in this area and important recent developments.

Reproducibility: Yes

Additional Feedback: NOTE (post rebuttal): I didn't change my review, as a matter of expediency. But, I appreciate the authors' acknowledgement of my comments and support the plan for addressing them. (By the way, I also think "generative model" is a pretty standard setting in this community. I guess a brief clarification wouldn't hurt, but I wouldn't suggest using space to delve into the setting more deeply.) "corner stones" -> "cornerstones" "the sample complexity of model based MARL algorithms has rarely been investigated": The Rmax paper was one of the first papers to study RL sample complexity AND dealt with MARL. Maybe it hasn't been "recently" investigated? Or revisited in light of newer approaches? "especially model-free ones, as the condition for guaranteeing convergence in the latter fails to hold in MARL": Convergence of model-free Q-learning variants in games has been known since the 90s. Maybe you are specifically talking about the setting where one agent can't see the action of the other? (No, that's definitely not the setting discussed. Although I'm still at a loss to understand the context of the comment.) "See a detailed description as follows." -> "A detailed description follows."? "Contribution": Are you following the NeurIPS style? I think subsections are supposed to be numbered. "[51] also focused": I think it's bad style to use the references as nouns. (It's definitely hard for me to read...) "has relatively limited literature" -> "has a relatively limited literature"? "to find eps-optimal policy" -> "to find an eps-optimal policy" "the state-action/Q-value function under (mu, nu) are defined" -> "the state-action/Q-value function under (mu, nu) is defined" "for the single-agent RL" -> "for single-agent RL"? "which are efficient in finding the NE of Ghat": They really only find eps-NEs, right? (I think this concept is described more clearly later, but it still might be best to be a bit more careful here.) "note that different from the single-agent setting," -> "note that, different from the single-agent setting," or "note that, in contrast to the single-agent setting," "smooth PLANNING ORACLE": Why the bold every time? I think bold (or, better, italics) the first time makes sense as it calls attention to defining a new term. After that, normal text would be appropriate. "an instance of the smooth Planning Oracle" -> "an instance of a smooth Planning Oracle"? "from the single-agent setting, to zero-sum MGs" -> "from the single-agent setting to zero-sum MGs" Some sloppiness in the bibliography: - "Princeton university press" -> "Princeton University Press" - "Pac bounds" -> "PAC bounds" - "markov games" -> "Markov games"


Review 3

Summary and Contributions: This paper studies the sample complexity of model-based MARL algorithms in two-player discounted zero-sum Markov games. Based on an auxiliary Markov game, the authors present the complexities in finding epsilon-approximate NE value and epsilon-approximate NE policy. Moreover, the authors introduced Smooth Planning Oracle to to improve the sample complexity of obtaining the NE policies, which can be can be near-minimax optimal.

Strengths: The paper gives the first near-minimax optimal sample complexity for model-based MARL in zero-sum Markov games with detailed proof. More specifically, the authors established the first near-minimax optimal sample complexity of both achieving the Nash equilibrium value and policy.

Weaknesses: The proof of main lemma 3.1 is mainly adopted from lower bound for MDPs proof in [4, 17], more clarification to highlight the difference would be good.

Correctness: The cliams and proofs of the paper appear to be correct.

Clarity: The paper is well-written and structured clearly. The contributions and proof roadmap are well summaried.

Relation to Prior Work: The aurthors has disccussed the connections and gaps to model-free, turned based and single agent optimality analysis. Also mentioned the difference between existing model-based MARL complexity analysis.

Reproducibility: Yes

Additional Feedback: It is a bit confused when first see the generative model in abstract and introduction, the formal definition has been given in line 129. Therefore, more specific description about the generative model in abstract or introduction would be good to quickly capture the point, maybe the generative model of state transition? Just curious, can policy guided sampling reduce the sampling complexity of model-based learning?


Review 4

Summary and Contributions: This paper provides theoretical results for zero-sum Markov games when a generative model is available. The authors show that a simple model-based algorithm, which builds an empirical MDP and applies a planning oracle, can obtain near minimax sample complexity.

Strengths: I recommend an acceptance for this paper. This paper provides strong theoretical results for the zero-sum Markov games. The authors show with a plug-in empirical MDP, (1) any planning oracle can find near optimal NE value, and (2) a soft planning oracle can find near optimal NE policy. Although some analysis techniques directly borrow from recent work on model-based RL [Agarwal et al., 2020], I believe the theoretical contributions are solid and nontrivial, as they need to carefully handle the minimax game setting.

Weaknesses: It might be helpful to give some discussion on why introducing the soft planning oracle and how it can improve over planning oracles in general.

Correctness: I briefly check the proofs in Appendix A and it looks correct to me.

Clarity: The paper is well-written and easy-to-follow. The authors also give enough helpful discussions. I particularly enjoy the discussion of lower bound in Appendix B.

Relation to Prior Work: The authors clearly discuss related works.

Reproducibility: Yes

Additional Feedback:

[Author Response · NeurIPS 2020]

We sincerely thank all the reviewers, and feel really honored to receive such positive and constructive comments. We are in the process of incorporating many of the changes into the final version.

**Reviewer 1   1)**   We will improve the broader impact section by emphasizing the implications of our theoretical results on applications. **2)**   We will mention total variation distance in the appendix, and correct the typo on "Corollary 3.3". Thanks for the careful reading.

**Reviewer 2   1)**   Note that the smooth planning oracle is not needed throughout the paper, and is thus not the "primary concept" in our paper. It is only used in Sec. 3.2 for finding $\epsilon$-NE policies with near-optimal sample complexity. We have justified this in lines 209-231, and will add some demonstration on simple games in the final version. **2)**   Yes, by saying "rarely", R-MAX is definitely one of the very few ones. We have discussed R-MAX in lines 82-83. We will change the wording in the final version. **3)**   By saying "especially model-free ones..." this sentence, we simply meant that the convergence techniques for single-agent Q-learning/model-free methods cannot be applied directly in MARL. The works on Q-learning in games you mentioned exactly conquered this issue, with non-trivial efforts. We will add more clarifications on this. **4)**   We will address all the grammatical comments/typos in the final version. Thanks a lot for the careful examining.

**Reviewer 3   1)**   We will add more comparisons with [4,17] on the lower bound proof, in the final version. **2)**   We thought the "generative model" setting is a standard one. Thanks for pointing this, and we will add the clarification. We believe an adaptive policy guided sampling may indeed decrease the sample complexity of "model-based" methods, if more structure about the "model" is known beforehand, e.g., which state transition is more significant. Otherwise, for general models without special structure, estimating the whole transition model (approximately) seems inevitable, and our matching upper-bound seems to be hardly improvable.

**Reviewer 4**   On the smooth planning oracle: First, technically, our "leave-one-out" proof technique requires some "smoothness" of the change of either the "value-function" or the "policy" (for proving $\epsilon$-approximate NE value and $\epsilon$-NE policy, respectively), when one state is made "absorbing". When showing the near-optimal sample complexity to find $\epsilon$-approximate NE value, the change in value function is indeed smooth, so no "smooth planning oracle" is needed; when showing the near-optimal sample complexity to find $\epsilon$-NE policy, the small change in the Markov game cannot guarantee the change of the "best-response" policy of the opponent to be small, too. In fact, a small change in the value may correspond to a drastic change in the best-response policy. Thus, we introduce the smooth planning oracle to address this. Second, computationally, as we have discussed in lines 209-231, such an oracle can be readily satisfied in practice, especially in the regularized setting. Regularized setting makes solving the matrix game a strongly-convex-strongly-concave problem, and thus improves the computational efficiency, over the un-regularized ones. Whether it is possible to achieve "near-optimal sample complexity" for achieving "$\epsilon$-NE policy" without the smooth oracle may require totally different proof techniques, and is left as our future work.

[Meta-Review · NeurIPS 2020]

All reviewers agree that this is a solid work that deserves publication. We encourage the authors to follow the reviewers' suggestions when preparing the camera-ready version of their paper.